# Cryopreservation and revival of Hawaiian stony corals using isochoric vitrification

Matthew J. Powell-Palm [1,2,8,9] ✉, E. Michael Henley [3,4,8] ✉,
Anthony N. Consiglio [5], Claire Lager[3,4], Brooke Chang[5], Riley Perry[3,4],
Kendall Fitzgerald[3], Jonathan Daly [6,7], Boris Rubinsky [5] & Mary Hagedorn[3,4,9]

Corals are under siege by both local and global threats, creating a worldwide reef crisis. Cryopreservation is an important intervention measure and a vital component of the modern coral conservation toolkit, but preservation techniques are currently limited to sensitive reproductive materials that can only be obtained a few nights per year during spawning. Here, we report the successful cryopreservation and revival of cm-scale coral fragments via mL-scale isochoric vitrification. We demonstrate coral viability at 24 h post-thaw using a calibrated oxygen-uptake respirometry technique, and further show that the method can be applied in a passive, electronics-free configuration. Finally, we detail a complete prototype coral cryopreservation pipeline, which provides a platform for essential next steps in modulating post-thaw stress and initiating long-term growth. These findings pave the way towards an approach that can be rapidly deployed around the world to secure the biological genetic diversity of our vanishing coral reefs.

Coral reefs around the world are imperiled by local and global threats, leading to shocking declines in species and genetic diversity[1]. Over the past five decades, coral reef losses have accelerated[2], and future losses are predicted at 70–99%. Coral reefs are some of the oldest, most diverse ecosystems on our planet and act as an oceanwide nursery for a quarter of all marine life[3]. They protect coastlines, cities, and homes; are critical to food security; and support widespread recreational activities[4,5]. Today, the economic value of the world's coral reefs is estimated at $10 trillion USD per year[6]. However, despite their unparalleled global economic and social value, coral reefs continue to decline, threatening most life on Earth, and more imminently the lives and livelihoods of resource-poor but reef-rich nations[7].

The coming decade will be critical for the future of coral reefs, and therefore broad scientific interventions are sought[8]. One such intervention action is cryopreservation. Cryopreserved coral sperm has been banked for more than a decade and has been used to assist gene flow both in the critically endangered Caribbean Elkhorn coral[9] and on the Great Barrier Reef[10], and recent advances in laser rewarming techniques have also enabled cryopreservation of coral larvae[11]. However, while preserved reproductive materials can be utilized for reef restoration and diversification, climate change and its related bleaching events are negatively impacting coral reproduction[12–14]. Specifically, certain species, when exposed to warming waters and the stress of subsequent bleaching, demonstrate a long-term loss of sperm motility, reduction of egg size, and abnormal larval development[12] or loss of reproduction the following years[15]. It is hypothesized that this multi-year impact may be due to ultraviolet radiation (UVR) damage incurred by reproductive stem cells during bleaching, when the coral loses the UVR protection produced by its symbionts[16]. Therefore, an approach that can circumvent these climate change-related

[1]J. Mike Walker '66 Department of Mechanical Engineering, Texas A&M University, College Station, TX, USA. [2]Department of Materials Science and Engineering, Texas A&M University, College Station, TX, USA. [3]Smithsonian National Zoo and Conservation Biology Institute, Front Royal, VA 22630, USA. [4]Hawai'i Institute of Marine Biology, University of Hawai'i at Mānoa, Kāneʻohe, HI 96744, USA. [5]Department of Mechanical Engineering, University of California Berkeley, Berkeley, CA, USA. [6]Taronga Institute of Science and Learning, Taronga Conservation Society Australia, Mosman, NSW 2088, Australia. [7]Centre for Ecosystem Science and Centre for Marine Science and Innovation, School of Biological, Earth and Environmental Sciences, University of New South Wales, Sydney, NSW 2052, Australia. [8]These authors contributed equally: Matthew J. Powell-Palm, E. Michael Henley. [9]These authors jointly supervised this work: Matthew J. Powell-Palm, Mary Hagedorn. ✉e-mail: powellpalm@tamu.edu; henleym@si.edu

reproduction issues is needed, enabling cryopreservation of the entire coral organism without waiting for increasingly uncertain yearly reproductive events.

Towards that goal, we describe herein the successful cryopreservation of whole cm-scale coral microfragments using a cutting-edge thermodynamic cryopreservation process called isochoric vitrification.

Scaling cryopreservation protocols from isolated cell suspensions (such as sperm) to large tissues or whole organisms presents a number of physical challenges, chief among them that these more complex systems (unlike their cell suspension counterparts) can generally tolerate little-to-no ice formation in the system. The growth of ice within complex tissues can cause both mechanical damage via its expansion during crystallization and osmotic damage via the solute rejection process. This therefore generally requires that they be cryopreserved in an ice-free "glassy" state, by the complex kinetic process of vitrification[17]. Vitrification is highly dependent on both system chemistry and cooling/warming rates[17,18], and as sample size increases, the decreasing surface area-to-volume ratio sharply limits both surface-driven heat transport and the mass transport of cryoprotectants moving into the tissue/water moving out of the tissue.

Recent work has provided a wellspring of new approaches that grapple with these scaling issues: optimized metallic meshes to enable ultrafast cooling rates in microscale samples, nanoparticle-enhanced optical and electromagnetic warming techniques to enable ultrafast and uniform warming, advances in perfusion technologies to enable loading of higher-osmolality, higher-viscosity cryoprotective solutions, etc.[19–22]. However, the many exciting solutions that have emerged in the past five years are unified by two common limitations: sample size (≪1 mL) and/or tolerance of these samples to the highly concentrated (7–10 M) and potentially toxic solutions required.

In this work, we present biological validation of isochoric vitrification[23], a cryotechnology based on the ice-growth-limiting principles of aqueous isochoric thermodynamics. We find that this technique can facilitate a vitrification process at low cooling and warming rates (<100 °C/min and <400 °C/min respectively) in bulk-volume samples (5 mL) of a relatively low-molarity (~4 M) vitrification solution. We combine this technique with key advances in coral husbandry and evaluation to achieve cryopreservation and revival of the endemic Hawaiian coral *Porites compressa*, showing cryopreservation of mature coral fragments of ~1 cm length scales. After cryopreservation, we demonstrate a 24 h post-thaw survival of coral microfragments using oxygen-uptake respirometry. Finally, we show that this technique can be successfully applied without active monitoring, in an electronics-free isochoric chamber ready for use in the field. We suggest that this technique can be applied any time of year and in most relevant environments and thus holds great promise for ensuring the maintenance of coral diversity.

## Results

### Isochoric vitrification: thermodynamic premise and validation

Vitrification techniques typically employ isobaric (constant-pressure) thermodynamic conditions, wherein the system (i.e., the biological sample of interest and surrounding preservation media) is open to the atmosphere or another reservoir of pressure[24]. Under these conditions, the uniquely large density difference between liquid water and ice-Ih does not appreciably affect the kinetic processes of ice nucleation and vitrification. However, if the system is rigidly confined and *denied* access to the atmospheric pressure reservoir, i.e., held under isochoric (constant-volume) conditions, both the equilibrium thermodynamics and the kinetics of nucleation and growth change dramatically[23,25–29]. Under confinement, the expansion exacts a sizable energetic toll on the nascent ice phase, which has been shown in theory and experiment to both limit the ultimate growth of ice[25,26] and reduce the probability of its initial nucleation from arbitrary

supercooled aqueous systems[27–29]. The closed and confined system also eliminates several other potential sources of nucleation, including interactions with air[28,30], liquid surface instabilities[27], and three-phase contact lines[31]. Avoiding the nucleation and growth of ice at temperatures between the melting point and the glass transition is the central challenge in facilitating vitrification, and we thus hypothesized that the same suite of physical effects driving ice avoidance in the past decade of high subzero isochoric experiments may also help to facilitate vitrification in isochoric systems[23].

In order to probe this supposition, we designed a custom isochoric vitrification chamber fitted with a pressure transducer (Fig. 1a) and leveraged the principle of pressure-based isochoric nucleation detection[28] to evaluate the isochoric vitrification behaviors of a new coral preservation solution. We based this solution, called CVS1 (1.05 M dimethyl sulfoxide (DMSO), 1.05 M glycerol, 1.05 M propylene glycol (PG), 0.85 M trehalose in phosphate buffered saline (PBS, 0.15 M); total molarity 4.15; more details in Supplementary Note 1), on a solution used previously to preserve coral larvae at microliter volumes with the assistance of laser rewarming[11]. While the bulk vitrification process is typically evaluated visually[32], the pressure generated by ice formation in isochoric systems provides an alternative tool by which to evaluate potential phase transitions within the chamber, with both the presence of a pressure spike and its magnitude providing valuable physical information.

Figure 1 demonstrates the thermodynamic evaluation process used to determine whether vitrification may have occurred within the sealed chamber. In brief, the isochoric chamber, shown in Fig. 1a, is plunged into a bath of liquid nitrogen (up to the connection mating it to the pressure transducer, which is itself rigorously insulated to keep the encased electronics above −50 °C at all times), allowed to reach steady state, then plunged into a water bath at 27 °C with submersible pumps providing turbulent mixing. The temperature and pressure are monitored throughout, with a rapid increase in pressure indicating growth of ice in the system[25,28]. This simple plunge method provided maximum cooling/warming rates (i.e., at the periphery of the liquid sample; see Supplementary Note 2) of 96 +/−6 °C/min and 387 +/− 55 °C/min, respectively, and did not produce the sharp increase in pressure indicative of ice nucleation (Fig. 1b). CVS1 was thus concluded to vitrify successfully under these conditions. In order to verify this initial conclusion and explore the sensitivity of the vitrification process, we also performed the same experiments at slower cooling and warming rates, achieved by plunging the chamber with varying degrees of insulation. As shown in Fig. 1c, reduced cooling rates elicited ice-driven pressurization upon cooling to approximately 65 MPa. Similarly, when the chamber was once again cooled at 96 °C/min but rewarmed more slowly, the same pressurization was observed (Fig. 1d). This cooling/warming rate dependence is the hallmark of vitrification— a kinetic process that depends sharply on the thermal history of the sample.

In order to confirm whether the observed behavior of the CVS1 solution was indeed a product of the employed isochoric thermodynamic conditions, as hypothesized, we repeated the trials shown in Fig. 1b with an unsealed chamber open to atmospheric pressure. The solution froze in all cases (indicated by significant expansion of the sample, as opposed to the significant contraction that accompanies vitrification[33]), confirming that isochoric confinement facilitates the observed process. Additional detailed description of these thermodynamic tests is included in Supplementary Note 3.

Within the isochoric system, it should also be noted that even at the mild cooling rates that do produce detectable ice growth (as in Fig. 1c), some degree of partial vitrification is likely occurring within the system, driven by the unique combination of pressure effects and solute ripening that is characteristic to the isochoric freezing process[34]. This is suggested by the local maximum in pressure within the plausible temperature vicinity of the glass transition, which implies

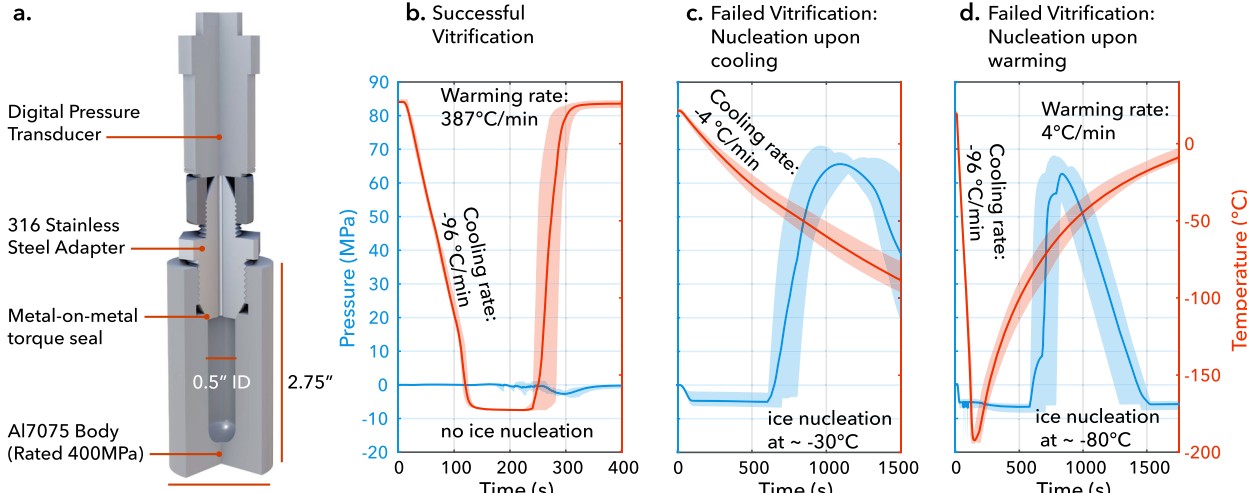

**Fig. 1 | Facilitating and verifying an isochoric vitrification process. a** Custom-designed isochoric chamber rated for 400 MPa/−200 °C. The chamber body is built of aluminum 7075-T651 for its combination of excellent thermal and mechanical properties, and the threaded adapter is built of 316 stainless steel. The adapter possesses a significantly lower temperature coefficient of thermal expansion compared to Al-7075, thus producing an advantageous thermal shrink-seal upon cooling. **b** Pressure–temperature–time (PTT) trace for successful vitrification, as indicated by the absence of a jump in pressure ($n = 10$ independent trials). **c**, **d** PTT traces for slower cooling (**c**) and warming (**d**) rates, each of which produce detectable ice nucleation and the according pressurization ($n = 10$ independent trials, each). Each vitrification trial reflected in (**b**–**d**) employed a fresh sample of CVS1 solution. Solid lines in (**b**–**d**) represent the mean PTT trace across trials, and shaded error bands represent the full range of results. Source data for (**b**–**d**) are provided as a Source data file.

a ceasing of ice growth and a gradual transition to the more contractile glassy state[33,35,36] (See Supplementary Notes 4, 5, and 6 for supporting calculations and further discussion). The notion of isochoric partial vitrification thus introduces the intriguing premise that a biological sample within the system could potentially still vitrify even while faced with ice growth elsewhere in the chamber.

However, this same competition between icy expansion and solution contraction, and the additional effects of contraction of the chamber, places limits on the absolute sensitivity of pressure-based vitrification detection, i.e., even in the case of Fig. 1b wherein no net pressure increase is observed, some amount of ice could still be forming. A first attempt at interrogating some of the intricacies of the thermo-volumetric effects at play was recently made by Solanki and Rabin[37], who produced a simplified heat transfer-mechanics COMSOL model studying a high concentration (7 molar) DMSO solution. While their first-order analysis was not yet able to account for the key physical processes needed to describe the isochoric vitrification process (e.g., tensioning of the liquid solution; contraction of the chamber itself; solution thermodynamics; ice nucleation kinetics; possible cavitation dynamics; etc.), it provided valuable insight supporting the notion that there exists a limit on the sensitivity of pressure-based vitrification detection that is a function of the complex contraction effects that may be present in a given solution. Their work also highlights the need for novel approaches to solution design for isochoric vitrification, which should aim to minimize solution thermal contraction. Untangling the complex material physics at play during the isochoric vitrification process presents an exciting field of future research and may benefit from further study at the material property level, the transport and kinetics level, and the protocol design level alike.

Finally, it should be noted that the ultimate extent to which vitrification occurs within the isochoric system can only be measured indirectly at present, by the likes of pressure monitoring, evaluation of the preserved biologic, etc. As such, the authors suggest that an immediate priority of the field may be to develop isochoric cryopreservation platforms that integrate x-ray or other penetrating optical evaluation methods to help provide vital direct observation of the isochoric cryopreservation process.

## Cryopreservation of Hawaiian stony coral fragments

Having identified a thermal protocol yielding no ice-pressurization of the CVS1 coral preservation solution under isochoric conditions, we proceeded to test this process on the Hawaiian coral *Porites compressa*. Our experimental workflow is shown in Fig. 2, and described in further detail below, in the "Methods," and in the SI.

## Selection and fragmentation

First, a healthy adult coral colony was located on the reef, from which a fragment approximately 10 cm × 20 cm was collected and maintained in an outdoor flow-through mesocosm system at the Hawaiʻi Institute of Marine Biology located in Kāneʻohe Bay, Hawaiʻi. This fragment was then further cut into several ~1 cm² "microfragments," each of which contained approximately 20 individual polyps, and glued to a gridded acrylic plate organized by genotype identification and date of microfragment processing. After a period of at minimum 10 and at maximum 30 days, the microfragments recovered fully from the stress of fragmentation and were ready for use in the cryopreservation process. Full recovery is here evaluated according to the standardized health metrics provided by Lager et al.[38] More information on the general practice of coral fragmentation and reef restoration from coral fragments is provided in the SI.

## Bleaching

The microfragments are brought into the laboratory and then bleached with a combination of menthol and light, according to the protocol of Lager et al.[38] (detailed also in Supplementary Note 7). This causes them to shed their algal symbionts, which in previous work and preliminary trials were found to interfere with the CPA-penetration/coral tissue dehydration process due to the differences in cryobiological parameters between the symbionts and the tissue[39,40]. Sufficient absence of symbionts post-bleaching is confirmed by pulse amplitude modulation (PAM) fluorometry.

## CPA loading

Bleached fragments (as shown in Fig. 2) are then introduced into full-concentration CVS1 in three submersion steps (33% CVS1, 2 min; 66%

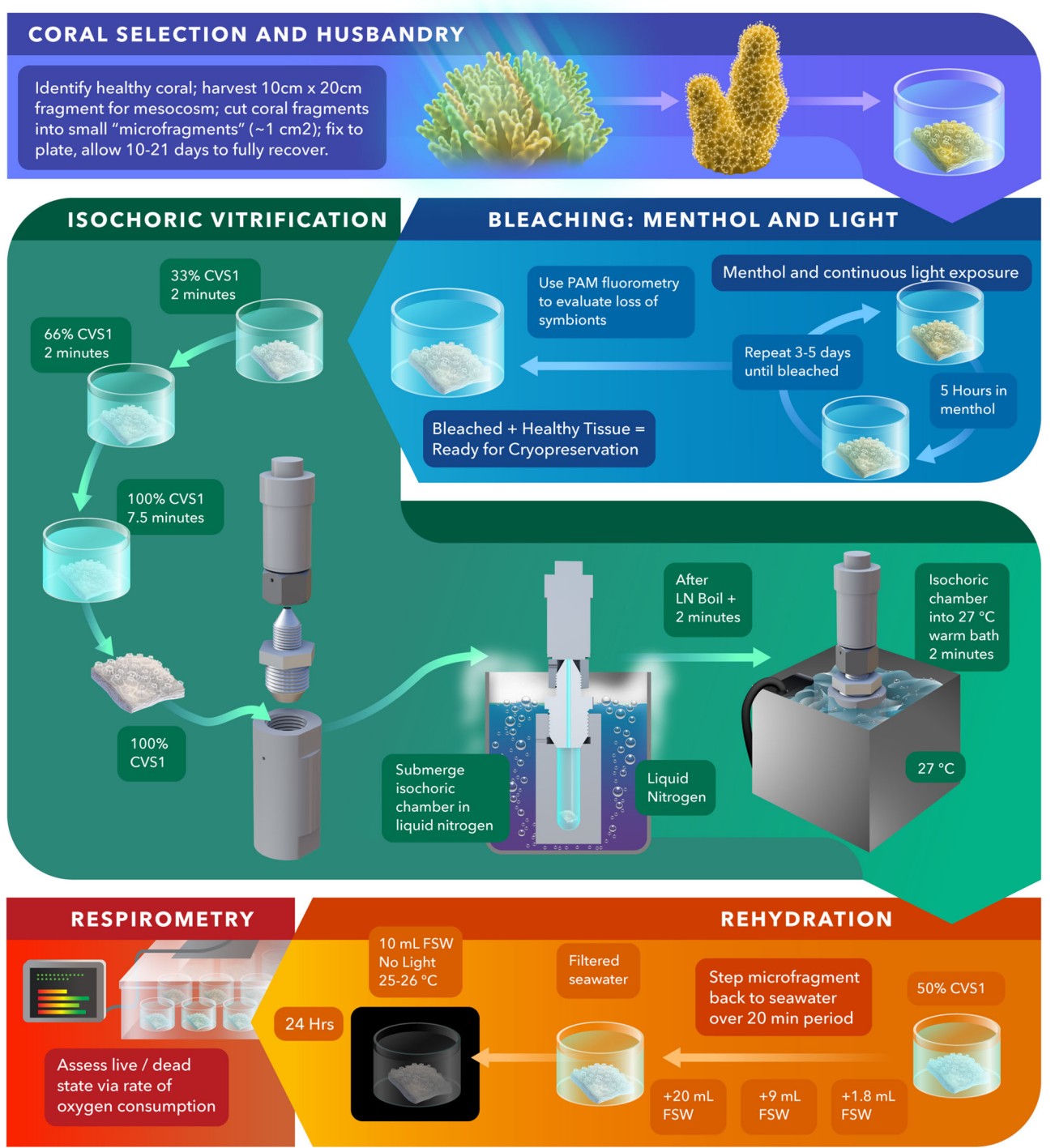

**Fig. 2 | Coral cryopreservation and revival protocol.** Each step shown here is described in brief in the text, and step-by-step procedures details are provided in the SI.

CVS1, 2 min, 100% CVS1, 3–7.5 min) in order to avoid significant osmotic stress. The composition of CVS1 was engineered to produce a combination of dehydration and CPA penetration, containing both non-penetrating extracellular agents (trehalose) and penetrating intracellular agents (DMSO, glycerol, and propylene glycol). While the ultimate degrees of penetration and dehydration achieved cannot be quantified here, we hypothesize that dehydration is the dominant process, as coral tissue contains extremely fast-acting water channels, characterized by their low membrane permeability[11,40]. Furthermore, in order to probe the sensitivity of the ensuing coral vitrification process to successful dehydration/penetration from CVS1 (and to screen for any acute toxicity effects), we introduce the diffusion time in the

full-strength solution as an experimental variable, testing exposure times of 3, 5, and 7.5 min.

### Isochoric vitrification of coral microfragments

Following CPA loading, a given coral microfragment is then placed into an isochoric chamber filled with CVS1 solution, sealed to a torque of 45 ft-lb, and promptly plunged into a bath of liquid nitrogen (LN$_2$). This process requires approximately 30 s, and care is taken to ensure that no bulk air bubbles are introduced into the chamber during loading, as the presence of bulk air has been shown to both alter the isochoric thermodynamic path[41] and aid in the nucleation of ice[30]. In our core suite of vitrification trials, the chamber was instrumented with a digital

pressure transducer (Ellison Sensors Inc., USA) as shown in Fig. 1a, and the exact protocol used to conduct the previous thermodynamic trials was also used for the biological trials, again employing the absence of an increase in pressure to confirm successful vitrification. The chamber is held in LN$_2$ until steady state is achieved (indicated by the cessation of LN$_2$ boiling). Two minutes thereafter, the chamber is transferred directly from the LN$_2$ bath to a circulating water bath at 27 +/− 1 °C, where it is submerged for two additional minutes. This temperature represents the maximum at which *Porites compressa* will not experience additional heat-based stress, thus eliminating the possibility of overheating. After warming is complete, the chamber is unsealed, and the coral is transferred into a petri dish of 50% CVS1 in filtered seawater. This dish is further diluted with filtered seawater (FSW) in several discreet steps (see Supplementary Note 7 for full details) over the next 19.5 min, at which point the coral is transferred to a fresh dish of 10 mL FSW and allowed to recover in darkness at 25–26 °C for the next 24 h.

### Respirometry technique for analyzing coral survival

Following experimental treatment, a quantitative technique by which to evaluate the health of coral fragments in real time was needed. While existing techniques are adequate in many circumstances, they fall short when external stressors (including thermal and chemical stressors) may elicit visible physiological responses in the coral that do not necessarily equate to cell death, tissue damage, or other long term deleterious effects. Our lab established the utilities and limits of some of these methods for *Porites compressa* in previous work[38], concluding ultimately that simple visual morphological evaluation, brightfield microscopic evaluation, and even confocal fluorescent evaluation provided limited use in establishing the relative health of stressed coral fragments for these experiments. In need of a more robust and quantitative method by which to evaluate the health of coral fragments, we used a microrespirometry technique to record oxygen consumption rates over time in order to differentiate between healthy and severely injured coral or dead coral[42].

Importantly, coral is considered a holobiont, comprised of coral tissues, algal symbionts (Symbiodiniaceae, which are removed by bleaching prior to vitrification), and a bacterial microbiome. We demonstrate here that the oxygen consumption balance between the bleached coral tissue and its bacterial microbiome provides a strong indicator of health. Generally, when coral tissue is healthy, the system can regulate potential overgrowth of bacteria and maintain a steady homeostasis. However, when a fragment is severely injured or dead, it is rapidly overrun and consumed by bacteria, a process fueled by a dramatic increase in oxygen consumption. In order to quantify this effect, we measured the 15-min oxygen consumption profiles of healthy bleached corals ($n = 59$ fragments from $N = 29$ genotypes; positive control) and cryo-injured corals ($n = 50$, $N = 24$; negative control), using a Loligo Systems temperature-controlled microplate respirometer (24 wells, 1700 μL volumes; Loligo Systems, Viborg, Denmark). The cryo-injured negative controls were produced by exposing the coral to precisely the same CPA-loading and cooling/warming protocol used during isochoric vitrification, and in precisely the same chamber, but without sealing the chamber, yielding a conventional isobaric thermodynamic state. As stated in the earlier thermodynamic description, the isobaric (unsealed) system yields rampant ice formation under identical thermochemical conditions, and this cryo-injury proved lethal to the corals. In preliminary trials, we also tested CPA-loading the fragments and plunging them directly into liquid nitrogen, without any surrounding media, but found that biological result was indecipherable from the isobaric protocols (both yielding significant visible ice formation), and thus proceeded using only the isobaric protocol for our cryo-injured negative controls.

The raw oxygen consumption profiles for all samples were normalized to 100% O$_2$ at time $t = 0$ and are shown in the first two panels of Fig. 3a. In order to facilitate comparison and statistical analysis, each profile was then fitted to an exponential function of the form $y = \exp(−a*t)$, characterized by the rate constant of oxygen consumption $a$ (all fitted rate constants are tabulated in Supplementary Table 4). The distribution of this rate parameter describes the variability within each experimental group. The median values for the experimental groups (positive and negative controls) are depicted as the aggregate-fit lines in Fig. 3b, with the shaded regions denoting the 25% and 75% quartiles of $a$. These results demonstrate the characteristically different oxygen consumption profiles accompanying healthy and injured tissue, which we used as bounding markers of health in all additional experiments. Further details on fitting and all forthcoming statistical analysis can be seen in the "Methods."

### Experimental pipeline

Having established the requisite husbandry to prepare mature coral for preservation and respirometry to evaluate their health in real-time thereafter, we designed an experimental pipeline to simultaneously probe for solution toxicity and cryo-injury in our model isochoric vitrification protocol. In order to ensure robust genotypic representation, for each genotype gathered, microfragments were dedicated to the following groups: positive control; negative control; toxicity trials; isochoric vitrification trials. In each of the toxicity and vitrification trials, three exposure times in full-strength CVS1 solution were tested (3, 5, and 7.5 min). The vitrification trials proceeded as described previously, and in the event that unintended nucleation was observed in the pressure signal (which was found typically to stem from issues related to the pressure transducer; discussed in later sections), the trial was marked and the fragment was segregated from those undergoing successful vitrification for the purposes of analysis. For the toxicity trials, the microfragments were transferred directly from the CPA loading stage shown in Fig. 2 to the rehydration stage, with the intention of isolating potentially deleterious chemical or dehydrative effects that may act independently of the cryopreservation process. Importantly, we reiterate that the negative controls were cryo-injured by exposing them to precisely the same protocol as used for the isochoric vitrification trials, except without sealing the isochoric chamber, providing a sharp additional testament to the importance of isochoric conditions in facilitating the vitrification process.

### Coral fragments survive isochoric vitrification with sufficient osmotic conditioning

Figure 3 showcases the aggregated respirometry profiles of these experiments, both in raw (a) and fitted (b) form, and the averaged oxygen consumption rate parameters '$a$' are shown in Fig. 3c. Pairwise two-sided two-sample $t$ tests were conducted for comparison between every group, and the resulting significance table is shown in Fig. 3d. Several important results emerge. First, all toxicity trials (labeled "tox" in Fig. 3) produced oxygen consumption profiles that are statistically indistinguishable from the positive bleached control, indicating that exposure to the CVS1 solution for any of the timepoints tested did not produce detectable damage on its own. Conversely, the vitrification trials showed a distinct dependence on CPA loading time; 3 min exposure ($N = 15$, $n = 24$, labeled "iso-3min") yielded respirometry profiles statistically similar to the negative controls ($p = 0.07$), indicating significant injury, and 5 min exposure ($N = 16$, $n = 28$, labeled "iso-5min") yielded profiles statistically different from both negative and positive controls ($p = 0.02$ and $p = 0.006$, respectively). Importantly, however, 7.5 min CPA exposure ($N = 18$, $n = 29$, labeled "iso-7.5 min") yielded respirometry profiles both statistically indecipherable from the healthy controls ($p = 0.11$) and different from the negative controls ($p = 0.004$), indicating successful cryopreservation of the fragment and survival of the coral. To our knowledge, this represents the first successful cryopreservation and revival of the mature coral organism.

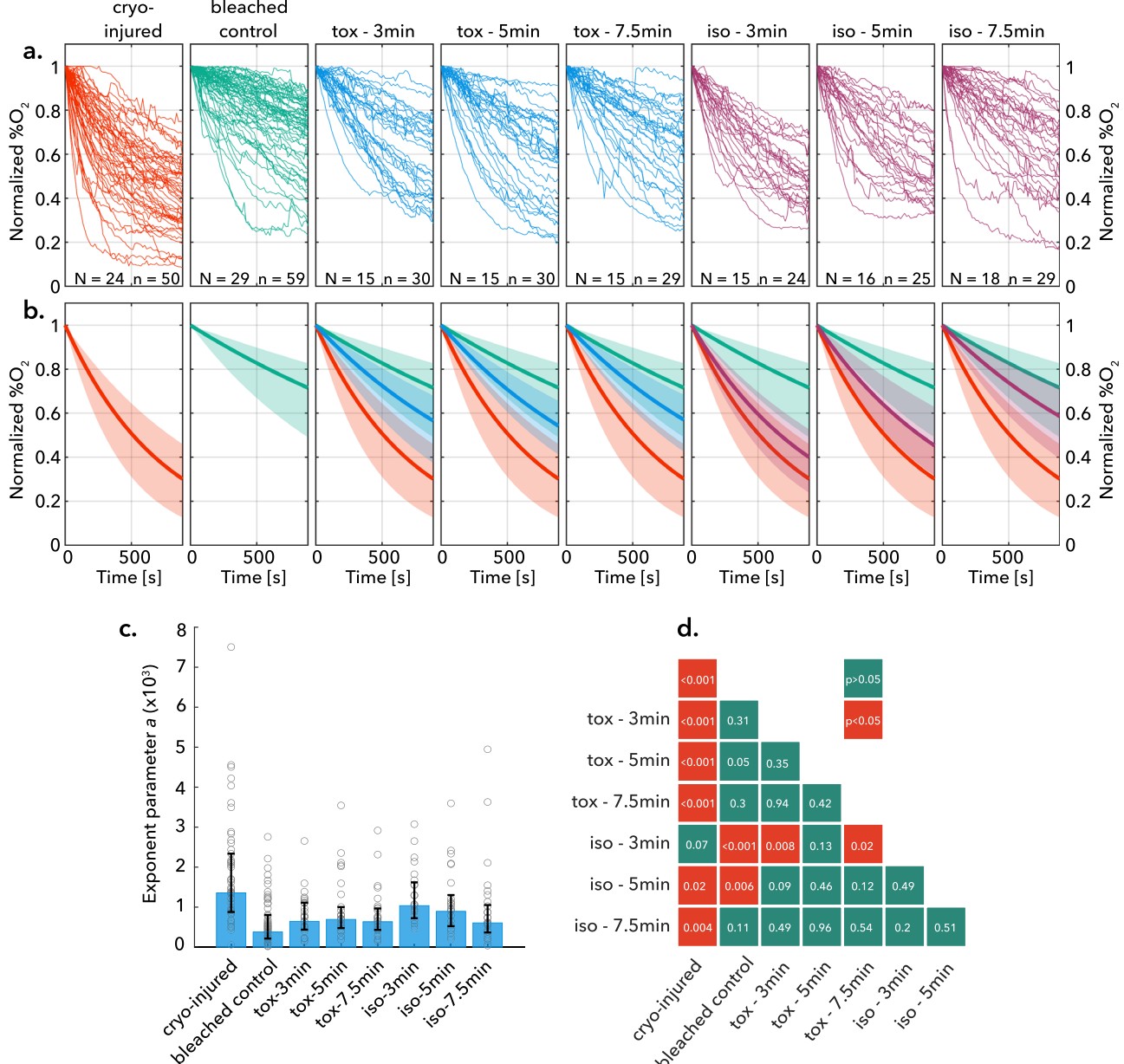

**Fig. 3 | Coral respirometry results. a** Raw and **b** exponentially fitted normalized oxygen consumption curves for four groups of data: positive control (live healthy bleached coral), negative control (cryo-injured coral exposed to ice formation via conducting the vitrification protocol in an unsealed chamber), CVS1 toxicity experiments (labeled "tox − $X$", where $X$ is the coral fragment equilibration time in full-strength CVS1 cryoprotective solution), and isochoric vitrification experiments (labeled "iso − $X$", where $X$ is the same). Number of genotypes $N$ and total samples $n$ in each data set are displayed at the boom of each panel in (**a**). Solid lines in (**b**) represent median values and shaded regions are bounded by 25%/75% quartiles of the fitted exponential rate parameter $a$, which is itself plotted in (**c**) histogram with medians and 25%/75% quartiles of the exponential fit parameter for each treatment. **d** provides a matrix of pairwise two-sided two-sample $t$ test results indicating significant difference between groups, demonstrating that isochoric vitrification after 7.5 min of osmotic equilibration in CVS1 produces coral that are statistically indistinguishable from healthy controls and statistically distinct from injured coral. $p$ values greater than 0.01 are shown rounded to two decimal places, and $p$ values less than 0.01 are shown rounded to three decimal places. Source data for all panels are provided as a Source data file, and exact $p$ values are provided therein.

It is key to note further that all the data shown for the vitrification trials in Fig. 3 reflect successful vitrification of the external carrier solution surrounding the fragment, as indicated by the absence of a rise in pressure. Thus, the improvement in coral health post-vitrification with increasing CPA loading time suggests that, while the external solution may be vitrifying, ice formation may have been occurring within the coral tissue when the loading period was insufficient, i.e., at less than 7.5 min of equilibration time in full CVS1. This provides a testament to the essential role of effective dehydration and CPA loading for vitrification protocols, and as new techniques are developed to quantify CPA penetration in complex tissues, the degree

of CPA penetration should be rigorously characterized. It should also be noted that, despite convective warming, no thermal stress cracking was observed in any samples.

## Developing an unmonitored, electronics-free cryobanking approach
Having identified a husbandry, loading, and vitrification pipeline that yielded surviving coral 24 h post-thaw, we sought to modify our protocol to maximize its readiness for applied banking efforts. In our preliminary thermodynamic testing and applied preservation testing, continuous monitoring of the pressure was required to ensure no

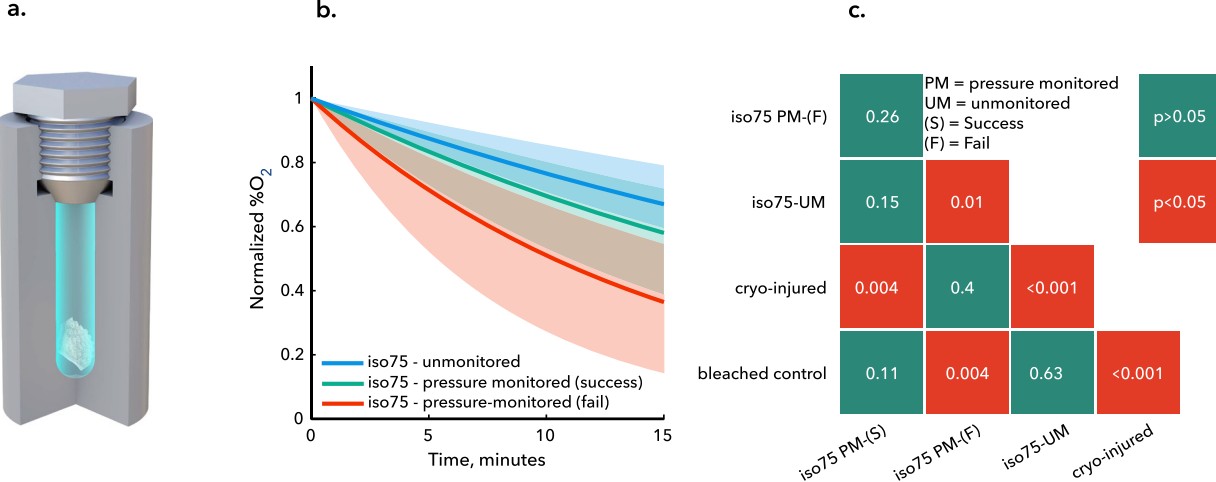

**Fig. 4 | Unmonitored isochoric vitrification for scalable cryobanking. a** Passive, field-ready isochoric chamber that does require active monitoring or electronics of any kind. **b** Aggregated median oxygen consumption curves for coral preserved via an unmonitored isochoric vitrification process vs. pressure-monitored equivalents deemed either successful or unsuccessful based on their pressure histories. Shaded regions are bounded by 25%/75% quartiles. **c** Matrix of pairwise two-sided two-sample *t* test results indicating significant difference between groups, showing that unmonitored samples are statistically indistinguishable from healthy bleached coral and statistically different from cryo-injured coral. **b**, **c** also indicate that the unmonitored chamber shown in (a) produces better vitrification results in total as compared to the monitored device shown in Fig. 1a. In (**c**), *p* values greater than 0.01 are shown rounded to two decimal places, and *p* values less than 0.01 are shown rounded to three decimal places. Source data for (**b**, **c**) are provided as a Source data file, and exact *p* values are provided therein.

erroneous ice nucleation had occurred. However, continuous sample-to-sample monitoring with a pressure transducer is implausible for a real-world banking effort. The high-fidelity transducers required not only add significantly to the cost of each preservation assembly, but also decrease the likelihood of successful vitrification, adding unnecessary additional surface area, material interfaces, opportunities for air entrainment, and thermal gradients to the system (all of which can help to stimulate ice nucleation)[30,31]. As such, in the final phase of our experiments, we eliminated the pressure transducer and all active monitoring/electronics from the system, in order to produce a more cryobank-ready preservation tool.

As shown in Fig. 4a, we replaced the pressure transducer and SS316 adapter with a solid Al7075 plug that employed the same metal-on-metal surface sealing mechanism and repeated our successful vitrification protocol (i.e., that with 7.5 min CPA equilibration time) on an additional set of coral microfragments (N = 7, n = 22). Leveraging the high statistical power of our respirometry results for injured coral, healthy coral, and coral deemed successfully vitrified by way of pressure verification, we then used the respirometry results of these 22 unmonitored trials to infer the success or failure of vitrification in the simple-capped system. As shown in Fig. 4b, c, the unmonitored trials (labeled "iso75-unmonitored") yielded respirometry profiles statistically indecipherable from both the successful pressure-monitored trials (labeled "iso75-pressure monitored (success)") and the control bleached trials (p = 0.15 and p = 0.63, respectively), suggesting both that this method yields continued successful vitrification and that this protocol may be used to the desired effect without any electronics or active monitoring.

It should be further noted that removal of the pressure transduction apparatus appears likely to have increased the success rate of vitrification by potentially reducing the number of unintended nucleation events. We infer this by also comparing in Fig. 4b the respirometry results of the pressure-monitored trials that failed to vitrify (n = 9 out of a total of 37 trials; labeled "iso75-pressure monitored (fail)"). The unmonitored results in the capped system are statistically identical to those for healthy coral, whereas the pressure-monitored results that show nucleation are not. This suggests that the rate of successful vitrification is increased in the capped system

relative to the measured 75.7% success rate in the aggregated pressure-monitored trials.

## Limitations of current data

The data presented in this work demonstrates a mL-scale isochoric vitrification process by which whole coral fragments may be cryopreserved, revived, and verified to be healthy at 24 h post-thaw. However, significant further development of recovery methods and husbandry techniques will be required to move incrementally toward full restored health, resumed calcification, and long-term growth.

Critically, coral exhibit a wide range of responses to environmental and biological stressors, and their recovery from these responses can take anywhere from days or weeks (for mild stressors such as our menthol-light bleaching process[38]) to months or years (for extreme stressors like prolonged exposure to warming oceanic conditions[12,15]). With this in mind, we chose the 24 h post-thaw evaluation time point in order to directly probe the efficacy of the cryopreservation process, at high throughput and independent of the long-term stress response. Now, armed with unambiguous proof that the coral survive the isochoric vitrification process itself, future work by our group and others must focus on developing methods to ensure that these preserved coral can fully recover from the inevitable stress of cycling some 230 °C in temperature. Preliminary observations from our group suggest that with no dedicated stress modulation and husbandry, cryopreserved coral will begin to succumb to bacterial degradation within approximately 3 days.

In particular, future efforts of our group and others will develop methods of actively combating bacterial degradation of the tissue during the recovery process, giving the tissue sufficient time to re-establish its holobiont equilibrium. Bacterial management via pro-, pre-, and anti-biotic treatments has emerged as an essential tool in the treatment of stressed coral, with recent data even demonstrating that it can combat Stony Coral Tissue Loss disease[43], and we hypothesize that anti-biotic intervention will be key to modulating any long-term post-preservation stress responses.

Other urgent topics of future research include potential effects of the cryopreservation process on the coral genome and microbiome; the efficacy of isochoric vitrification for other species of coral, which

may not only have different biological responses but different propensities for stimulating ice nucleation based on their surface properties; and the ease of translation of isochoric vitrification into field conditions.

## Potential impacts of cryoconservation on the future of coral reefs

New models of future oceans that use an average temperature elevation of 1.5 °C[44–46] suggest that only 2.5–5% of the worlds reefs may remain by the mid-2030s, restricted to two areas of the world. Given the limited scope of global efforts being made to mitigate heating of our oceans, the Intergovernmental Panel on Climate Change (IPCC) suggests that a future target of 2.0 °C and above is a more likely future scenario[46], leaving a mere decade before the diversity of the world's coral reefs may be all but lost.

Time is of the essence for coral reefs, and therefore we must take immediate, concrete steps to develop a wide suite of conservation practices with concomitant and robust ex situ processes. These processes include maintaining coral species in captivity[47], creating coral mesocosm breeding facilities to facilitate and improve reliable breeding[48], and enhancing existing biorepositories to include coral sperm, larvae, symbionts[10], and coral fragments at critical global nodes[49].

For ex situ cryostorage, sexually produced sperm and larvae are ideal given their size and assortment of genes, but this material is only available a few days per year and only in remote locations around the globe. Moreover, climate change and other stressors are increasingly and negatively impacting coral reproduction[12,15] and, as a consequence, the physiological robustness of this produced sexual material.

If successfully transitioned into a restoration practice, isochoric vitrification could be applied strategically around the globe in relatively non-impacted areas hundreds of days per year, thereby quickly securing the genetic diversity and biodiversity of the whole coral organism using this field-friendly technology. Although coral microfragments are bleached before the isochoric process, they can be re-infected during the grow out process, and mature colonies can then be placed into ex situ reproduction facilities to yield sexually-produced offspring or transitioned to existing coral nurseries[50–52].

Ex situ processes will have a tremendous role to play in the future of maintaining coral reefs, and best practices mandate that these be applied side-by-side with current reef-based restoration practices that focus largely on in situ fragmentation and outplanting to current or former reefs. Isochoric vitrification provides a versatile example of one such process, and efforts to ready it for field applications should be undertaken with the utmost urgency.

## Methods

All corals used in this study were collected under permits SAP 2022-22 and SAP 2023-31, which were approved and issued by the State of Hawaiʻi Department of Land and Natural Resources.

### Chamber design and fabrication

The chambers shown in Figs. 1a and 4a were custom designed and custom CNC machined from round-stock Aluminum 7075-T651, then finished with a type-II anodize to prevent corrosion in the presence of aqueous salt solutions. This metal was chosen for its exceptional combination of high strength and high thermal conductivity, which significantly outclasses steels and titanium alloys. The wall thickness was selected to ensure that it could withstand internal pressures in excess of 400 MPa, with approximately 300 MPa being the highest pressure observable from an aqueous isochoric freezing process and much lower pressures expected. We note that these chambers may easily be extended to arbitrary lengths without appreciably affecting the stress profiles in the walls or the thermal profiles during processing, as both of these are functions of the radial dimensions. The

pressure transducers used (ESI GD4200-USB-4000-DE) have a pressure detection rating up to 400 MPa, and feature a standard Autoclave F-250-C female high-pressure adapter fitting. If other researchers are interested in designing, fabricating, or acquiring similar chambers for isochoric testing, guidance is available from the corresponding authors.

### Thermodynamic tests

In order to explore the thermodynamic premise of isochoric vitrification, chambers loaded with the CVS1 solution were cooled and warmed at different rates while the temperature and pressure of the system were monitored.

**Chamber assembly and test preparation.** First, teflon tape was applied to both threaded sides of the 316 SS adapter, which was then torqued into the ESI pressure transducer (rated for 0.1–400 MPa) to 27 ft-lbs. The CVS1 solution was loaded slowly into the 5.55 mL AL7075 chamber volume via syringe to avoid the formation of bubbles. The chamber was filled until the solution formed a convex meniscus over the top opening, then the adapter volume (0.22 mL) was filled with solution until a convex meniscus formed over the hole. The mated pressure sensor and adapter were torqued into the chamber to 45 ft-lbs, and excess solution was allowed to escape through the weephole. A type T thermocouple (OMEGA) was insulated with a layer of Kapton tape and inserted into the chamber weephole to monitor temperature throughout the experiment. The digital pressure transducer (Ellison Sensors Inc.) was wrapped in insulating bubble wrap to maintain the temperature above −50 °C. A water warming bath fitted with two submersible water pumps (VIVOSUN) to provide turbulent mixing was warmed to 27 °C. ESI-USB software and Omega logging software were used to record the pressure and temperature, respectively.

**Isochoric vitrification tests.** To achieve maximum cooling rates (96 +/−6°C/min), the chamber was submerged up to the bottom of the pressure sensor throughout the process. Once nitrogen boiling ceased (indicating steady state), the chamber remained submerged for an additional 2 min before being rapidly moved to the warming bath. For maximum warming rates (387 +/− 55 °C/min, $n = 10$), the chamber was submerged in the warming bath directly between the two pumps. The reduced cooling rates shown in Fig. 1c (4 +/− 0.3 °C/min, $n = 10$) were achieved by insulating the chamber with a plastic bag (P4M 4×6, GT Zip) before submerging in the liquid nitrogen. The reduced warming rates shown in Fig. 1d (4 +/− 0.1 °C/min, $n = 10$) were achieved by removing the chamber from liquid nitrogen and allowing to cool in the air (at approximately 17 +/− 0.7 °C). Between experiments, the chamber and adapter volumes were rinsed with ethanol and vacuumed to remove any remaining liquid or particles.

**Isobaric ice formation test.** To compare the effects of isobaric conditions on the CVS1 solution to those observed in the isochoric case, an unsealed chamber was used. The open AL7075 chamber was carefully filled with CVS1 solution up to the edge of the cylindrical chamber volume (not including the threaded region). The weephole of the chamber was sealed with Kapton tape to prevent the pooling of liquid nitrogen into the chamber. The chamber was then submerged up to the weephole in liquid nitrogen. The chamber was observed while it remained in the liquid nitrogen bath to avoid condensation on the surface. Ice growth was evaluated visually based on significant expansion of the solution.

### Coral colony selection and husbandry

From June to December 2022, small (approximately 10 cm × 20 cm) colony fragments of *Porites compressa* were selected from patch reef sites across Kāneʻohe Bay, Oʻahu, Hawaiʻi and held in flow-through seawater mesocosms at the Hawaiʻi Institute of Marine Biology. Shortly

after collection, colonies were processed into 0.5 cm² microfragments (Page et al.[50]). Microfragments were allowed to heal and recover for at least 10 days prior to use in experiments and were not used beyond 30 days after microfragmentation. Corals were collected under Special Activity Permit Numbers SAP 2022-22 and SAP 2023-31 issued by the State of Hawai'i Department of Land and Natural Resources.

## Microfragment preparation

Due to differences in cryopreservation sensitivity between coral tissue and symbionts (Symbiodiniaceae require more time for cryoprotectant equilibration and faster freezing rates[39]), corals were bleached according to the protocol of Lager et al.[38]. After 3–5 days of menthol and light exposure to induce bleaching, microfragments for each genotype were visually inspected under a dissecting microscope for tissue integrity and overall coloration. To ensure loss of symbionts, corals were then assessed for any photosynthetic activity (Yield) using an Imaging PAM (Heinz Walz GmbH, Germany). Microfragments that had visible tissue recession or photosynthetic yield values greater than 0.1 were not used in experiments. After visual and photosynthetic yield assessments, corals for each genotype were then assigned a treatment designation: bleached (live) control, dead (bleached negative control), toxicity controls (3, 5, and 7.5 min), and Isochoric (3, 5, and 7.5 min); the time designations corresponded to the length of time in the dehydrating cryoprotectant solution for both toxicity controls and cryopreserved specimens.

## Experimental treatments

Isochoric vitrification experiments and CVS1 toxicity experiments began with the same procedure: portions of CVS1 were diluted with 0.22 micron filtered seawater (FSW) to 33% (10 mL FSW, 5 mL CVS1) and 66% (5 mL FSW, 10 mL CVS1) strength. Coral microfragments were then stepped into CVS1 at increasing concentrations: two minutes at 33%, two minutes at 66%, and 3, 5, or 7.5 min at 100% CVS1 depending on the experimental treatment designation. The isochoric chambers were filled with CVS1 per the same protocol used in the thermodynamic testing.

For toxicity trials, after completion of CPA equilibration, fragments were moved directly to the rehydration steps described below. For isochoric vitrification trials, fragments were placed using forceps into an isochoric chamber, taking care not to introduce any errant air bubbles.

Pressure-monitored vitrification trials then proceeded precisely as described in the thermodynamic testing section above. The unmonitored vitrification trials shown in Fig. 4 were performed on a subset of 7.5-min corals, with the only difference in procedure being the replacement of the pressure transducer with an identically threaded solid cap.

After warming, the chamber was promptly opened and the microfragment was placed in a petri dish with 1 mL of 50% CVS1 for 30 s to begin rehydration. For the remainder of rehydration, FSW was added to the dish with increasing volumes over time: 1.8 mL after the initial 30 s in 50% CVS1, 3 mL after another 30 s, 9 mL after 1 min, 18 mL after 3 min, and 20 mL after another 3 min. Toxicity controls and negative (cryo-injured) controls went through the same process of incremental dehydration in CVS1 and incremental rehydration.

Toxicity controls were not exposed to liquid nitrogen or warm bath. Negative controls were placed in an isochoric chamber and plunged in nitrogen according to the same procedure as described for the vitrification, but without sealing the chamber (with pressure sensor or solid cap), ensuring ice formation throughout the chamber.

## Respirometry measurements

After isochoric vitrification, corals from all treatments were placed into new 6-well plates in 10 mL of 0.22 micron FSW and maintained at 26 °C with no light. After approximately 24 h, a Loligo Microplate

Respirometry system (Loligo Systems, Viborg, Denmark) was used to measure oxygen consumption as a proxy for coral health post-thaw. A small aquarium air pump was used to aerate a bottle of FSW to fill the 1.7 mL wells of the glass microplate with 26 °C oxygenated seawater for hydration of the microplate sensors, calibration of the unit (at high oxygen concentration), and experimental measurements. For low oxygen concentration calibration, 1 g of sodium sulfite was dissolved in 50 mL of deionized water and allowed to equilibrate for a few hours at 26 °C as oxygen-free water. For the respirometry measurements, aerated FSW and corals were placed in the wells of the microplate, and the microplate was sealed with PCR film, silicone pad, and compression block (provided in the Loligo Systems kit). Care was taken to ensure that no bubbles were sealed in the wells before starting the trial, and measurements were taken in 15 s intervals for 15 min. In addition to the sealed microplate and sensor, the respirometry system was assembled with a small aquarium pump and heater to circulate 26 °C FSW around the microplate in an acrylic chamber to maintain temperature for the duration of the assessments. The microplate wells were rinsed with deionized water between assessments.

## Respirometry data analysis

Respirometry measurements generated time series data of the oxygen concentration within each individual microplate well. These measurements were repeated on ≥20 samples for each experimental group. In order to draw conclusions from the variable responses, a data reduction pipeline was developed. First, the oxygen concentration measurements (%$O_2$) were normalized to the initial concentration. Second, an exponential curve of the form $y = \exp(-a*t)$ was fit to the initial 5 min of normalized data using the *fit()* function in MATLAB 2022b, enabling each sample to be represented by a single exponential fit parameter $a$, and individual experimental groups to be represented by a distribution of the same. Statistical analysis of the respirometry data was conducted in order to compare the relative health of different experimental groups (e.g., cryo-injured, bleached control, toxicity, post-vitrification, etc.). The distributions of fitted exponential rate parameters computed for each group were compared using a two-sided two-sample $t$ test in MATLAB 2022b (*ttest2()* function). Effects are reported for significance levels of $p < 0.05$. A table of all exponential fit parameters is included in the SI, along with goodness-of-fit metrics for each experimental group.

## Reporting summary

Further information on research design is available in the Nature Portfolio Reporting Summary linked to this article.

## Data availability

All data described in this work are tabulated in the Source data file provided with this paper. If any assistance is required in accessing or interpreting this data, the reader is welcome to contact M.J.P.-P. (powellpalm@tamu.edu), with a response to be expected within 2–4 weeks. Source data are provided with this paper.

## Code availability

Two MATLAB scripts are provided in the Supplementary Information (Supplementary Code 1), which facilitate the exponential fitting of the raw respirometry data and the statistical analyses of the resulting fit parameters. If any assistance is required in accessing or interpreting this code, the reader is welcome to contact M.J.P.-P. (powellpalm@tamu.edu), with a response to be expected within 2–4 weeks.

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

## Acknowledgements

This work was generously funded across institutions by a grant from the Revive & Restore Catalyst Science fund (to M.H., M.J.P.-P., B.R.). The Berkeley and Texas A&M contingents recognize additional support from NSF ERC ATP-BIO under NSF EEC Grant No. 1941543 (to M.J.P.-P., B.R.). A.N.C. was supported by an NSF Graduate Research Fellowship under Grant No. DGE 1752814. The authors thank Andy Grams for graphic design assistance.

## Author contributions

M.J.P.-P. and M.H. conceived and supervised the study as a whole and contributed to all aspects. B.R. conceived the premise of isochoric vitrification. E.M.H., C.L., R.P., and K.F. carried out all experiments involving coral, with input from J.D. and M.J.P.-P. and supervision from M.H. B.C. performed all thermodynamic experiments, with support from A.N.C. and supervision from M.J.P.-P. and B.R. A.N.C. performed the statistical analysis. M.J.P.-P., E.M.H., M.H., B.C., and A.N.C. wrote the paper, and all authors reviewed the manuscript and provided critical feedback.

## Competing interests

B.R. has filed a 2017 patent application related to isochoric vitrification, which is under review as of the date of submission of this work. M.J.P.-P. and B.R. have financial stakes in a commercial entity that holds the license to the said patent application. The other authors declare no competing interests.
