## [Peer Review File · Nature Communications]

Cryopreservation and revival of Hawaiian stony corals using isochoric vitrificationREVIEWER COMMENTS

Reviewer #1 (Remarks to the Author):

This paper describes a study using isochoric vitrification to cryopreserve coral reef tissue. The method is based on applying pressure on a solution containing samples of interest to avoid ice formation (causing sample damage) during subsequent cooling. The advantage of this method (a variation of isochoric freezing) is that possibly lower CPA concentrations and lower cooling rates can be used to achieve vitrification. The authors show proof of principle that cryosurvival can be achieved using this method. Sample viability was based on oxygen consumption measurements before and after processing.

The application of this type of advanced and state of the art cryotechnology provides some exciting and promising results that can possibly be used to preserve endangered coral reef species, which was hitherto extremely difficult. It is needless to say that biobanking of such species is becoming increasingly important to combat biodiversity decline.

The paper is well written, and nicely illustrated with schematic figures and data. However, several major and minor comments need to be addressed.

Major points:

1.) The equation used for fitting the respiration data does not accurately describe the oxygen consumption process. The lag that is seen in the beginning is not captured by an exponential decay function. I would like to urge the authors to use a different function for fitting the respiration data that more accurately describes the oxygen consumption process. Maybe a modified form of a sigmoid or logistic function can be used (something like $f(t) = 1/(1+e^{-kt})$). How much sample (g) is needed for the measurements? Is this needed for the calculations?

2.) Provide a statement/discussion on a recent paper noting intrinsic difficulties with isochoric vitrification: Cryobiology 111 (2023) 9–15, Is isochoric vitrification feasible?

3.) What is the evidence that samples are truly completely vitrified? I don't think vitrification can be claimed without for example x-ray data.

Minor points:

1.) Page 2....and the osmotic action of chemical cryoprotectants. It is not entirely clear what is meant here. I would say something like: .. mass transfer limitations of cryoprotectants moving into the tissue and water moving out of the tissue.

2.) Page 3...a novel thermodynamic technique, isochoric vitrification...

The phrasing 'thermodynamic technique' is rather vague and awkward. I would rephrase and elaborate what is meant, e.g.: ...isochoric vitrification, a novel type of cryotechnology based on the thermodynamic principle of freezing point depression (or avoidance in this case) induced by applying pressure on a solution.

3.) ...a combination of dehydration (driven by trehalose, which cannot penetrate the coral tissue) and....

The remark that dehydration is driven by trehalose is not entirely accurate. In vitrification, permeable CPAs greatly contribute to dehydration because water moves faster out of tissues/cells than CPAs move in. The cooling process in vitrification is not started when all CPAs have moved into the sample (as in slow freezing) but when a sample is significantly dehydrated. Trehalose (sucrose is probably just as good in this case, and cheaper), is particularly used as osmotic stabilizer during rewarming when CPAs in the tissue cause swelling upon return to isotonic conditions.

4.) In the supplementary file, the model of Couchman and Karasz (the T_g of a blend/mixture is the average of the T_g of its molecular components) is used to provide an estimation of the glass transition temperature of the vitrification solution. I am not convinced that the T_g of anhydrous trehalose can be included in the computations, because in contrast with the other compounds

(DMSO, glycerol, PG) trehalose is a solid at room temperature. Can the authors find literature support (or experimental evidence) that the estimated T_g is ~-100C?

Reviewer #2 (Remarks to the Author):

Thanks for the opportunity to review the manuscript 'Cryopreservation and revival of Hawaiian stony corals via isochoric vitrification' (NCOMMS-23-09180). This study demonstrated the feasibility of using a novel thermodynamic technique, namely isochoric vitrification, for cryopreservation of mature coral fragments of 1-cm length scales. This was for the first time that coral fragments showed viability after cryopreservation in this scale in size, indicating the potential for use of this new technology for coral conservation to combat a global crisis of coral decline.

Overall, the manuscript is well written. The problems and background are well introduced, the methods are rigorously described, the results are clearly presented, and the conclusions and discussion are soundly supported by the data and existing literature. A major problem of the work is the duration of the post-thaw examination for 24 hr. Although this is strong support for the feasibility, there is doubt regarding whether the team has done the examinations beyond the 24 hr but chose not to show the extended examination data. Please see my comments below to help improve the quality of the manuscript.

Major:

1) Did the authors investigate the post-thaw viability of microfragments after 24 hr. If this has been done, data need to be added. It will not decrease the quality of the manuscript if no viability is shown after 24 hr, but it is an important piece of information for other researchers to develop further studies. Additional experiments may be needed if such an examination has not been done.

2) Has any work been done in vitrification of microfragments with similar treatments mentioned in the present work without the use of isochoric vitrification? The manuscript mentioned the use of cryo-injured microfragments as a negative control. Please provide the details regarding how this negative control was produced. In addition, this is a negative control for the respirometry data but not a negative control for the use of the isochoric chamber. In a previous experiment, an unsealed chamber open to atmospheric pressure was used as a comparison. Similarly, vitrifying microfragment samples processed with the same conditions by use of classic vitrification (direct plunging to liquid nitrogen) would be a necessary control or comparison. This experiment may need to be added.

3) A few points about the hardware can be further explained. What is the cost of making such a chamber? How was the chamber fabricated? How to reproduce the chamber if other researchers intend to repeat the work? Can this chamber be metal-3D printed? Can the chamber be made bigger in the future to accommodate high-scale throughput?

4) Why did the present work choose to preserve the microfragments instead of larvae? It seems existing publications have established adequate knowledge about coral larvae. Could the isochoric vitrification be used for larvae cryopreservation?

5) It seems with the volume of the chamber multiple pieces of fragments can be loaded. Has this been tested?

6) A fundamental assumption for the usefulness of this novel technology is that coral can be restored or reestablished in the wild with microfragments after being bleached as described in the manuscript. Has this been done? How many species have demonstrated the feasibility and how easy is the process? This needs to be further explained and more references are needed.

Minor:

Lines 63-67: It was stated that 'However, while preserved reproductive materials can be utilized for reef restoration and diversification, climate change and its related bleaching events are

negatively impacting coral reproduction. Therefore, a climate change-resistant approach is needed to enable cryopreservation of the entire coral organism without waiting for increasingly uncertain yearly reproductive events.' Please specify how climate change affects coral reproduction and the application of cryopreservation for sperm and larvae. Based on the description, it is assumed that the collection of gamete samples and larvae can be very sensitive to climate change. If this is the case, it needs to be pointed out more clearly. Also, a 'climate change-resistant approach' is overstated. The present work introduced an alternative strategy for coral cryopreservation, but nothing is climate change-resistant. The process of sample collection and eventually restoration still needs to deal with climate change with this novel approach.

Lines 106-108: Need a reference here.

Line 204: What is the definition of 'recover fully'?

Figures 3 and 4: Describe the hr post-thaw for all data presented.

Reference: Thorough proofreading of the reference list is needed. There are many errors (e.g., species name not italicized)

Reviewer #3 (Remarks to the Author):

Key Results

Powell-Palm et al. developed and tested a methodology to cryopreserve small microfragments of a living coral colony. They used a pressure sensor device embedded within the cryopreservation equipment to show when vitrification occurred successfully, tested a series of methods within two components of the cryopreservation process to optimize success, and then showed that cryopreserved specimens were successfully revived using respirometry. Overall, this is an incredible breakthrough for cryopreservation science and coral conservation.

Validity

Although I had a hard time understanding some of the initial validation sections of the paper, the authors did a nice job using language that can be relatively easy to digest. The step by step figures aided in the understanding of the validation of vitrification process.

There could be more introduction into why little to no ice formation is critical to success (Page 2 lines 73/73) and exactly what vitrification is within the context of coral cryopreservation.

The respiration technique was a nice way to show that the corals were alive and functioning, but the increase in oxygen consumption associated with bacterial blooms in dying corals was difficult to digest a bit without a reference or comparison data (Page 8, line 270). These statements suggest a rapid increase in oxygen consumption when a coral dies, but bleached controls do not show this...but the injured cryo frags do. Perhaps that is what should be emphasize more or maybe I am just missing something.

Data and Methodology

All seems available within the text and supplemental documents

Analytical Approach

All analyses appear appropriate

Suggested Improvements

See comments within other sections of the review.

Clarity and Context

Page 2, line 60: change to 'sperm has been banked'

Page 2, line 83: change to 'warming, and advances in perfusion...'

Page 3, line 109: what is 'ice-lh'?

Page 3, line 114: change to 'in theory and experimentally...'

Figure 1. I couldn't quite figure out how the ice nucleation temperature was determined (~-30C for c and ~-80C for d)

Page 6, 'Selection and fragmentation' section should be written in past tense

Figure 2 - just wanted to give a shout out that this is a great figure

Page 7, line 221: identify what DMSO and PG are within the text

It seems that there are two variables that are experimentally tested: 1 is the diffusion time of the

solution and I can't really figure out what the other variable is. Explain more. Is the diffusion time the 'tox' in Figure 3? If so, what exactly is the 'iso' variable. I am having a hard time deciphering this part.

May want to check that the red and green blocks and lines within the Figures are appropriately accessible for colorblind readers.

Nature Communications: Response to Reviewers: "Cryopreservation and revival of Hawaiian stony corals via isochoric vitrification"

We would like to thank all three reviewers for taking the time to provide insightful and helpful reviews. We have incorporated all of their suggestions into the revised text, which we feel is dramatically strengthened as a result of their input. Accordingly, we have credited their contributions as "Anonymous Reviewers" in the Acknowledgements section.

For the ease of the editor and reviewers, this document provides a thorough point-by-point response to each of the reviewers' concerns, comments, and questions. In the text that follows, the original text of the reviews is provided in **blue**; the authors responses are provided in **black**; and any verbatim text copied from the revised manuscript is **highlighted**. Additionally, in the revised text document, all significant changes (small language corrections not included) are **highlighted**.

The revised text has evolved significantly from its starting point in response to these reviews, and we hope that our changes have duly addressed all of the reviewers' concerns. If not, we would be happy to make further revisions as deemed necessary for publication in *Nature Communications*.

Reviewer #1 (Remarks to the Author):

This paper describes a study using isochoric vitrification to cryopreserve coral reef tissue. The method is based on applying pressure on a solution containing samples of interest to avoid ice formation (causing sample damage) during subsequent cooling. The advantage of this method (a variation of isochoric freezing) is that possibly lower CPA concentrations and lower cooling rates can be used to achieve vitrification. The authors show proof of principle that cryosurvival can be achieved using this method. Sample viability was based on oxygen consumption measurements before and after processing.

We thank the reviewer for their close reading of the manuscript and the excellent ensuing review. However, before we proceed point by point, we would like to take this opportunity to address one aspect of their interpretation that is not consistent with the text (no doubt due to our own failures of communication!). Namely, the method used here is **not** based on applying pressure, and is in fact based on **avoiding** the application of pressure. Indeed in conventional isochoric *freezing*, the thermodynamic goal of the technique is to allow some portion of the system to crystallize in order to pressurize the remainder of the system, which will remain in a liquid state due to the freezing-point-depression from said pressurization.

However, it is essential to highlight that the purpose of vitrification protocols of any nature is of course to **avoid** ice formation— which in an isochoric system also means avoiding pressure! In

isochoric systems, because it is ice formation that **produces** positive pressure; the avoidance of ice formation must necessarily also be accompanied by the avoidance of a pressure increase.

Indeed, this is the approach we have taken in this manuscript – we demonstrate that with the CPA solution of interest, if cooled sufficiently quickly, mass ice formation (and the according pressures) can be avoided. This is what we aim to convey in Figure 1; Figure 1b shows that we can plunge the chamber to LN2 temperatures and rewarm in a convective water bath **without** pressurization, indicating minimal formation of ice. In Figures 1c and 1d we show that if this same system is cooled slowly or warmed slowly, pressure **does** emerge, indicating extensive ice growth and the hallmark thermal-rate-dependence of vitrification processes.

We will comment on this further in response to the reviewer's other comments, but want to clarify at the outset that we do **not** expose the coral to high pressures during this preservation process. We sincerely apologize for not conveying this clearly enough in the original submission, and will work to highlight this aspect in our revision.

The application of this type of advanced and state of the art cryotechnology provides some exciting and promising results that can possibly be used to preserve endangered coral reef species, which was hitherto extremely difficult. It is needless to say that biobanking of such species is becoming increasingly important to combat biodiversity decline.

The paper is well written, and nicely illustrated with schematic figures and data.

We thank the reviewer for their kind words here, and agree wholeheartedly on the importance of the biobanking of threatened species.

However, several major and minor comments need to be addressed.

Major points:

1.) The equation used for fitting the respiration data does not accurately describe the oxygen consumption process. The lag that is seen in the beginning is not captured by an exponential decay function. I would like to urge the authors to use a different function for fitting the respiration data that more accurately describes the oxygen consumption process. Maybe a modified form of a sigmoid or logistic function can be used (something like $f(t)=1/(1+e^{-kt})$). How much sample (g) is needed for the measurements? Is this needed for the calculations?

We thank the reviewer for this comment. It may be difficult to see in Figure 3 because of the large number of distinct curves represented, but generally there is not an initial lag (though this does happen in occasional outliers). When selecting the fitting function, a large set of potential functions were evaluated for suitability. The exponential function was found to provide the best fit based on a standard R^2 coefficient-of-determination fit metric calculated in MATLAB. We had not included this original goodness-of-fit information in the initial submission, but we have now added a table to the SI showing that the single-parameter exponential fit provides average R^2 values of 0.7 - 0.92 across all experimental groups (Supplementary Note S8). We also tested the logistic function for good measure, but found R^2 values $\ll 0.5$ for every group.

Regarding the weight of the samples, we standardize them based on fragment dimensions rather than weight, because of the unique biological composition of the coral. Stony coral, unlike mammals, are comprised principally (by weight) of skeleton, which contributes a vast majority of the weight of a given sample but does not metabolize, respire, or contribute other active biological processes. The living coral holobiont exists as a thin “skin” on this skeleton, and it is thus not meaningful to attempt to normalize by weight of the sample. This is another reason why it is essential to use high numbers of individual samples. In Figure 3 for example, 279 different coral fragments are represented.

Based on our goodness-of-fit checks and robust sample numbers, we believe the experimental and analytical approach we’ve used for the respirometry herein captures well the biophysical distinction between cryo-injured and healthy coral that we are looking for. Looking towards more nuanced oxygen consumption-based health analyses that may be applied during future study of the prolonged recovery process, we are also actively investigating more advanced respirometry data analysis methods that may consider the heterogeneous composition of the sample as well as higher frequency dynamical signatures.

2.) Provide a statement/discussion on a recent paper noting intrinsic difficulties with isochoric vitrification: Cryobiology 111 (2023) 9–15, Is isochoric vitrification feasible?

We thank the reviewer sincerely for mentioning this – indeed we had already intended to cite this paper in any revised draft (it was published after our initial submission). We already had text in the original manuscript describing the thermal contraction / potential cavity formation effects that Solanki & Rabin identify, but we have now expanded that discussion incorporating their work explicitly (highlighted in revised manuscript). We feel this expanded discussion (which also blends into discussion additions based on the reviewer’s next comment below) greatly strengthens the early thermodynamic section of the paper. The new expanded ending to the thermo discussion is copied below:

“However, this same competition between icy expansion and solution contraction, and the additional effects of contraction of the chamber, places limits on the absolute sensitivity of pressure-based vitrification detection, i.e., even in the case of Fig. 1b wherein no net pressure increase is observed, some amount of ice could still be forming. A first attempt at interrogating some of the intricacies of the thermo-volumetric effects at play was recently made by Solanki & Rabin [38], who produced a simplified heat transfer-mechanics COMSOL model studying a high concentration (7 molar) DMSO solution. While their first-order analysis was not yet able to account for the key physical processes needed to describe the isochoric vitrification process (e.g. tensioning of the liquid solution; contraction of the chamber itself; solution thermodynamics; ice nucleation kinetics; possible cavitation dynamics; etc.), it provided valuable insight supporting the notion that there exists a limit on the sensitivity of pressure-based vitrification detection that is a function of the complex contraction effects that may be present in a given solution. Their work also highlights the need for novel approaches to solution design for isochoric vitrification, which should optimize minimization of solution thermal contraction.

Untangling the complex material physics at play during the isochoric vitrification process presents an exciting field of future research and may benefit from further study at the material property level, the transport and kinetics level, and the protocol design level alike.

Finally, it should be noted that the ultimate extent to which vitrification occurs within the isochoric system can only be measured indirectly at present, by the likes of pressure monitoring, evaluation of the preserved biologic, etc. As such, the authors suggest that an immediate priority of the field may be to develop isochoric cryopreservation platforms that integrate x-ray or other penetrating optical evaluation methods to help provide vital direct observation of the isochoric cryopreservation process.”

3.) What is the evidence that samples are truly completely vitrified? I don't think vitrification can be claimed without for example x-ray data.

This is an excellent and difficult question, and one the authors have labored extensively to balance. The reviewer is absolutely correct that it cannot be directly empirically verified that the liquid mass within the chamber (and furthermore within the sample) is *completely* vitrified. We mentioned this in original manuscript, and the possibility is now discussed at further length, incorporating the work of Solanki & Rabin. Initially, we suggested within the manuscript that vitrification was the appropriate description because:

- A) No positive pressure was read in our “successful vitrification” protocol, indicating that, at the least, any potential ice produced was considerably less than ice growth experienced under slower freezing conditions, which, per Figure 1.c, produces ~65 MPa of pressure.
- B) Speaking generally, if the sugar-CPA system does **not** freeze to a large degree (and produce the according pressure), it will vitrify when it passes its glass transition temperature (which, while not known precisely, is limited by the glass transition temperature of pure water, which is well above the temperature of LN2).
- C) Again speaking generally, if some limited ice forms in the system, to an extent yet undetectable by the pressure sensor, that ice will reject solutes into the remaining solution, thereby ripening the remaining liquid and increasing its likelihood of vitrification.
- D) When our solution is plunged into nitrogen in the isochoric chamber, but with the chamber unsealed (providing standard atmospheric pressure conditions), the solution is observed both to freeze and to accordingly expand significantly, as visible in the photos in the SI. When the chamber is sealed, net expansion of the interior contents (relative to their initial room temperature state) obviously cannot happen without the production of pressure – thus verifying that what is happening under isochoric conditions is certainly not what is happening under isobaric conditions.

- E) Most importantly, the coral preserved under isochoric conditions emerge from the process healthy, while the coral preserved under identical thermal / chemical / protocol conditions but with the chamber open are critically injured by ice formation.

--

This combination of logical points, which are argued with more brevity in the original manuscript, is what leads us to conclude that what must be happening within the chamber when no positive pressure from expansion of ice is detected is vitrification. We will note further that in the vitrification of complex biological matter in ANY situation, there is always an element of this physical reasoning.

Conventional vitrification for example is often (though certainly not always) evaluated by simple visual inspection – but the observer of course cannot peer into the *center* of a “vitrified” rat liver or a rabbit heart or a live coral, and thus cannot know for certain *prima facie* that the interior of the biologic has perfectly vitrified. To make matters worse, the center of the biologic is the point of the system that will experience the lowest cooling rate, and is thus MOST likely to form ice!

In order to overcome this physical uncertainty, vitrification trials of large biological matter are paired with biological evaluations (both based on functionality and on post-mortem histological or structural evaluation), which provide an additional indirect indication that ice has not formed, and that vitrification was successful. In our case, we tailored our biological evaluations to emphasize the negative biological effect of ice formation: our negative controls (“cryo-injured” in Fig 3) are simply coral fragments submerged in precisely the same solution as used in the isochoric trial, and in the same chamber, exposed to the same cooling and warming protocols, but **without** sealing the chamber and imparting the isochoric conditions. These coral are critically injured coming out of the cryopreservation process, while those under isochoric conditions are not. This biological comparison thus provides the final logical marker that leads us to claim that vitrification is happening.

However, the reviewer is absolutely correct that we cannot *know definitively* that the entire system is *completely* vitrifying without the use of x-ray or similar, even if we can estimate with high confidence that the system is certainly vitrifying to some extent. We have called out this “indirect measurement” vs “direct observation” distinction at the end of our updated thermodynamic discussion.

The authors humbly suggest that the distinction of pure vitrification vs. partial vitrification, which is discussed in the revised manuscript (now at significantly increased length and incorporating the findings of Solanki & Rabin), should be interrogated at significant further length in the future as new physical understanding and experimental techniques emerge, but should not here prove disqualifying, based on the positive biological result.

We will close this piece of commentary by saying that we strongly defer to the wisdom of the reviewer and the editor on this point, and would be happy to update the language additionally, as the central finding of the paper of course is not the purity of vitrification, but instead the positive biological outcomes resulting from this cryopreservation process.

Minor points:

1.) Page 2....and the osmotic action of chemical cryoprotectants. It is not entirely clear what is meant here. I would say something like: .. mass transfer limitations of cryoprotectants moving into the tissue and water moving out of the tissue.

We thank the reviewer for flagging this, and have changed the text to align with their suggestion:

“Vitrification is highly dependent on both system chemistry and cooling/warming rates [18,19], and as sample size increases, the decreasing surface area-to-volume ratio sharply limits both surface-driven heat transport and the mass transport of cryoprotectants moving into the tissue / water moving out of the tissue.”

2.) Page 3...a novel thermodynamic technique, isochoric vitrification...

The phrasing ‘thermodynamic technique’ is rather vague and awkward. I would rephrase and elaborate what is meant, e.g.: ...isochoric vitrification, a novel type of cryotechnology based on the thermodynamic principle of freezing point depression (or avoidance in this case) induced by applying pressure on a solution.

We thank the reviewer again for helping us to clarify this language. We respectfully reiterate that the technique does not actually employ pressure-based freezing point depression, so we will not incorporate the precise language suggested, but we have revised that part of the text to give more detail and reduce the vagueness of the original draft. It now reads as follows:

“In this work, we present the first biological validation of isochoric vitrification, a new cryotechnology based on the ice-growth-limiting principles of aqueous isochoric thermodynamics.”

3.) ...a combination of dehydration (driven by trehalose, which cannot penetrate the coral tissue) and.....

The remark that dehydration is driven by trehalose is not entirely accurate. In vitrification, permeable CPAs greatly contribute to dehydration because water moves faster out of tissues/cells than CPAs move in. The cooling process in vitrification is not started when all CPAs have moved into the sample (as in slow freezing) but when a sample is significantly dehydrated. Trehalose (sucrose is probably just as good in this case, and cheaper), is particularly used as osmotic stabilizer during rewarming when CPAs in the tissue cause swelling upon return to isotonic conditions.

This is an excellent point, and we have amended the text to reflect that all of the solutes in the solution indeed contribute to the dehydration process. Thanks again! It now reads more clearly:

“The composition of CVS1 was engineered to produce a combination of dehydration and CPA penetration, containing both non-penetrating extracellular agents (trehalose) and penetrating intracellular agents (DMSO, glycerol, and propylene glycol).”

4.) In the supplementary file, the model of Couchman and Karasz (the T_g of a blend/mixture is the average of the T_g of its molecular components) is used to provide an estimation of the glass transition temperature of the vitrification solution. I am not convinced that the T_g of anhydrous trehalose can be included in the computations, because in contrast with the other compounds (DMSO, glycerol, PG) trehalose is a solid at room temperature. Can the authors find literature support (or experimental evidence) that the estimated T_g is $\sim -100^\circ\text{C}$?

We thank the reviewer for a very interesting thought. We do not know of a fundamental thermodynamic reason why a given constituent should be excluded from the CK model based on its phase at room temperature, though we would certainly be intrigued to hear about one! Perhaps more importantly for the calculation at hand, our approach here is well supported in the literature and has been shown in both aqueous solutions of trehalose and other solutes of interest to cryobiology to agree relatively well with experiment. We refer the reviewer to works by Katkov & Levine (<https://www.sciencedirect.com/science/article/pii/S0011224004000884>) and by Adam Higgins' group (<https://journals.plos.org/plosone/article?id=10.1371/journal.pone.0190713>) for other examples of such calculations. We have also added these references to the relevant section of the SI to aid the reader.

Reviewer #2 (Remarks to the Author):

Thanks for the opportunity to review the manuscript 'Cryopreservation and revival of Hawaiian stony corals via isochoric vitrification' (NCOMMS-23-09180). This study demonstrated the feasibility of using a novel thermodynamic technique, namely isochoric vitrification, for cryopreservation of mature coral fragments of 1-cm length scales. This was for the first time that coral fragments showed viability after cryopreservation in this scale in size, indicating the potential for use of this new technology for coral conservation to combat a global crisis of coral decline.

Overall, the manuscript is well written. The problems and background are well introduced, the methods are rigorously described, the results are clearly presented, and the conclusions and discussion are soundly supported by the data and existing literature.

We thank the reviewer for taking the time for a thoughtful review, and agree with their synopsis and conclusions.

A major problem of the work is the duration of the post-thaw examination for 24 hr. Although this is strong support for the feasibility, there is doubt regarding whether the team has done the examinations beyond the 24 hr but chose not to show the extended examination data. Please see my comments below to help improve the quality of the manuscript.

This is a very good point, and the reviewer is correct that we did not sufficiently detail our logical process in choosing this time point. We have remedied that in the revised manuscript, which indeed much improves the quality. See details below.

Major:

1) Did the authors investigate the post-thaw viability of microfragments after 24 hr. If this has been done, data need to be added. It will not decrease the quality of the manuscript if no viability is shown after 24 hr, but it is an important piece of information for other researchers to develop further studies. Additional experiments may be needed if such an examination has not been done.

We thank the author for this line of questioning, which is indeed crucial to both interpretation and directing future studies. In retrospect, the discussion of our logic for selecting the 24h time point for evaluation was underdeveloped in the original text, a fact we have remedied in the revision.

To answer the reviewer's question directly, we do **not** yet have robust quantitative data on the time-evolution of the preserved coral's health at time points past 24h, but we **have** observed, preliminarily, that if no stress-modulation or active husbandry actions are taken, bacterial takeover (evaluated visually) appears to begin consuming these coral within approximately ~3 days post-preservation. We have added this point (and many accompanying points) to the manuscript for clarity.

Allow us to give a bit more background on the logic behind the 24h time point evaluation: Before the work presented here, cryopreservation of whole cm-scale coral fragments had never been achieved, and preservation of even microliter-scale single-polyp coral samples required use of a complex laser nanowarming platform (Daly et al., 2023). Thus, from a study design perspective, our priority was to establish scientifically that we can get large adult coral to survive a simple and fieldable cryopreservation process.

In order to demonstrate this, there were a large number of complicated variables to navigate, thermodynamic and biological, all of which may affect both the outcome and each other. At a high level, the most concerning of these were the vitrifiability of the solution, its potential biochemical toxicity, and the coupled dependence of these two factors on loading/diffusion time, which are what we focused on in this work.

From our preceding exploratory work on these topics, which we assembled into a companion paper submitted last fall (Lager et al., 2023), we clarified that the timeline to evaluate full long-term recovery of the coral (via conventional evaluation methods) after exposure to even very **mild** stressors (such as our benign bleaching technique or mild chilling) was **~10s of days**. As we were herein trying to rapidly establish the efficacy of the cryopreservation protocol, and not the evolution of the long term stress response, we needed an alternative evaluation. This drove us to establish a real-time quantitative respirometry technique, which we then applied at 24h post-thaw in order to establish survival of the coral, and facilitate direct comparison to the negative biological outcomes that occur if a conventional cryopreservation protocol (i.e. attempted vitrification in an open chamber) is used.

Thus, our goal was to de-couple the cryo-survival of the coral from the management of their long-term stress response. We have clarified this substantially in the text, greatly expanding the “Limitations of Current Data” section both to clarify the thought process, include our preliminary observations that preserved coral will begin to succumb to bacterial overload after approximately 3 days if no stress management is employed, and identify necessary next steps to manage the long-term stress response (which we are just starting on ourselves). The new sections reads as follows (also highlighted in the revised text):

“Limitations of current data

The data presented in this work demonstrates a mL-scale isochoric vitrification process by which whole coral fragments may be cryopreserved, revived, and verified to be healthy at 24h post-thaw. However, significant further development of recovery methods and husbandry techniques will be required to move incrementally toward full restored health, resumed calcification, and long-term growth.

Critically, coral exhibit a wide range of responses to environmental and biological stressors, and their recovery from these responses can take anywhere from days or weeks (for mild stressors such as our menthol-light bleaching process [39]) to months or years (for extreme stressors like

prolonged exposure to warming oceanic conditions [13,16]). With this in mind, we chose the 24h post-thaw evaluation time point in order to directly probe the efficacy of the cryopreservation process, at high throughput and independent of the long-term stress response. Now, armed with unambiguous proof that the coral survive the isochoric vitrification process itself, future work by our group and others must focus on developing methods to ensure that these preserved coral can fully recover from the inevitable stress of cycling some 230°C in temperature. Preliminary observations from our group suggest that with no dedicated stress modulation and husbandry, cryopreserved coral will begin to succumb to bacterial degradation within approximately 3 days.

In particular, future efforts of our group and others will develop methods of actively combatting bacterial degradation of the tissue during the recovery process, giving the tissue sufficient time to re-establish its holobiont equilibrium. Bacterial management via pro-, pre-, and anti-biotic treatments has emerged as an essential tool in the treatment of stressed coral, with recent data even demonstrating that it can combat Stony Coral Tissue Loss disease [44], and we hypothesize that anti-biotic intervention will be key to modulating any long-term post-preservation stress responses.

Other urgent topics of future research include potential effects of the cryopreservation process on the coral genome and microbiome; the efficacy of isochoric vitrification for other species of coral, which may not only have different biological responses but different propensities for stimulating ice nucleation based on their surface properties; and the ease of translation of isochoric vitrification into field conditions.”

— — —

We very much hope that this expanded discussion satisfies the reviewer’s concerns on this front. If they think that further / different information must be included, we will happily defer to their wisdom and make any further changes.

As we speak we are gathering data on the bacterial signatures that emerge on the stressed coral several days post-thaw, and are working with the Ushijima Lab of UNC-Wilmington to identify and test appropriate anti-biotic treatments. We have high hopes that with the right post-thaw husbandry and treatment, we will be able to nurse these coral into a full recovery.

References:

[Lager et al., 2023]

Metrics of Coral Microfragment Viability

Claire Lager, Riley Perry, Jonathan Daly, Christopher Page, Mindy Mizobe, Jessica Bouwmeester, Anthony N. Consiglio, Matthew J. Powell-Palm, Mary Hagedorn
bioRxiv 2023.01.03.522625; doi: <https://doi.org/10.1101/2023.01.03.522625>

[Daly et al., 2023]

The first proof of concept demonstration of nanowarming in coral tissue
Jonathan Daly, Jessica Bouwmeester, Riley Perry, Chris Page, Kanav Khosla, Joseph Kangas,
Claire Lager, Katherine Hardy, John C. Bischof, Mary Hagedorn
bioRxiv 2023.03.16.533048; doi: <https://doi.org/10.1101/2023.03.16.533048>

[Ushijima et al., 2023]

Ushijima, B., Gunasekera, S.P., Meyer, J.L. et al. Chemical and genomic characterization of a potential probiotic treatment for stony coral tissue loss disease. *Commun Biol* 6, 248 (2023).
<https://doi.org/10.1038/s42003-023-04590-y>

2) Has any work been done in vitrification of microfragments with similar treatments mentioned in the present work without the use of isochoric vitrification? The manuscript mentioned the use of cryo-injured microfragments as a negative control. Please provide the details regarding how this negative control was produced. In addition, this is a negative control for the respirometry data but not a negative control for the use of the isochoric chamber. In a previous experiment, an unsealed chamber open to atmospheric pressure was used as a comparison. Similarly, vitrifying microfragment samples processed with the same conditions by use of classic vitrification (direct plunging to liquid nitrogen) would be a necessary control or comparison. This experiment may need to be added.

We thank the reviewer for this careful reasoning, and agree completely. Gratefully, all of the experiments the reviewer has listed here **are already present** in the text – we apologize that our writing was not sufficiently clear to convey this, and have revised significantly to emphasize this fact.

To elaborate – indeed the non-isochoric vitrification control is hugely important to the interpretation. The negative control we employ is in fact just that! To produce the critically cryo-injured negative controls, we simply expose the coral to **precisely** the same protocol as under isochoric vitrification, but do not seal the chamber, yielding a standard isobaric/conventional thermodynamic system. This is precisely the same test we did to compare the thermodynamic effects in the earlier section, but now featuring the coral.

Importantly, trying to vitrify the coral in exactly the same solution under exactly the same conditions, but without the isochoric feature, produces dead or dying coral. In early preliminary experiments we also tried simply plunging the CPA-treated coral directly into nitrogen, which yielded the same negative result. Thus we proceeded with using the isobaric (unsealed) equivalent of our isochoric protocol as our negative control.

We discussed this briefly in the original manuscript, but in our revision we strongly emphasize that indeed the negative control *is provided by* attempting conventional vitrification.

To give the reviewer a bit of additional perspective as well, we respectfully refer them to the work of (Daly et al., 2023) – here, other members of our extended group worked to vitrify ~1

microliter droplets containing single-polyp coral microfragments. Under conventional atmospheric conditions, we were only able to successfully revive these (tiny!) fragments using a costly laser nanowarming technique; using conventional convective warming, these fragments will die. This underlines the extreme difficulty in vitrifying coral at **any** size, and underlines why we are so thrilled about the performance thus far of isochoric vitrification.

References:

[Daly et al., 2023]

The first proof of concept demonstration of nanowarming in coral tissue

Jonathan Daly, Jessica Bouwmeester, Riley Perry, Chris Page, Kanav Khosla, Joseph Kangas, Claire Lager, Katherine Hardy, John C. Bischof, Mary Hagedorn

bioRxiv 2023.03.16.533048; doi: <https://doi.org/10.1101/2023.03.16.533048>

3) A few points about the hardware can be further explained. What is the cost of making such a chamber? How was the chamber fabricated? How to reproduce the chamber if other researchers intend to repeat the work? Can this chamber be metal-3D printed? Can the chamber be made bigger in the future to accommodate high-scale throughput?

We thank the reviewer for these questions, and have added a section to the methods entitled “Chamber Design and Fabrication” (highlighted in the revised text and copied below). We cannot rigorously comment on the cost to build these devices, as they were custom built by our team and not built at any kind of relevant economy of scale (batch size of course hugely effects the per-unit cost of machined parts). We would be glad to help other academic researchers build their own however chambers, and have put a statement in the new section to that effect. Regarding fabrication method, we would not recommend 3D printing the device, as it should be designed for the maximum possible pressures encountered during isochoric processes, which are upwards of 300 MPa. We have added a comment specifying that the chambers were fabricated via CNC from solid stock. The chamber could easily be made longer or shorter, as the cooling and warming rates will remain consistent with consistent radial dimensions; increasing the radius would require additional analysis to screen the effect of a changing cooling/warming rate. We have also added a comment to this effect in the text.

“Chamber Design and Fabrication”

The chambers shown in Figures 1.a and 4.a were custom designed and custom CNC machined from round-stock Aluminum 7075-T651, then finished with a type-II anodize to prevent corrosion in the presence of aqueous salt solutions. This metal was chosen for its exceptional combination of high strength and high thermal conductivity, which significantly outclasses steels and titanium alloys. The wall thickness was selected to ensure that it could withstand internal pressures in excess of 400 MPa, with approximately 300 MPa being the highest pressure observable from an aqueous isochoric freezing process and much lower pressures expected.

We note that these chambers may easily be extended to arbitrary lengths without appreciably affecting the stress profiles in the walls or the thermal profiles during processing, as both of these are functions of the radial dimensions. The pressure transducers used (ESI GD4200-USB-4000-DE) have a pressure detection rating up to 400 MPa, and feature a standard Autoclave F-250-C female high-pressure adapter fitting. If other researchers are interested in designing, fabricating, or acquiring similar chambers for isochoric testing, guidance is available from the corresponding authors.”

4) Why did the present work choose to preserve the microfragments instead of larvae? It seems existing publications have established adequate knowledge about coral larvae. Could the isochoric vitrification be used for larvae cryopreservation?

We thank the reviewer for another good question. Our broader group has successfully cryopreserved coral larvae using laser warming (Daly et al. 2018, reference 12 in the revised text), but as stated in response #1 above, laser warming, while an effective scientific proof-of-concept, is **not** a useful restoration tool. Furthermore, there are several constraints that coral reproduction and the coral larvae themselves impose that make them much less preferable for isochoric vitrification (and conservation/restoration efforts more broadly). First, a coral species spawns for roughly two nights per year, so there exists a very limited time window for acquisition and testing; Second, we believe the cryo-physiological sensitivity of coral larvae would not adapt easily to the isochoric process. Coral larvae were successfully vitrified only by removal of cellular water and the presence of vitrification solution outside the larvae— virtually no cryoprotectant permeated the tissue. Coral tissues have water channels which allow water to move very quickly through the tissue (within seconds). Therefore, the *Fungia* larvae we vitrified in 2018 tolerated the dehydration of the vitrification solution for only ~30 sec, and past that the vitrification solution became toxic. This fast pace would not work well in the isochoric process because even at our fastest, it takes about 2 min to load and close the vitrification chamber prior to LN-plunging. Nevertheless, in exploratory tests for a different study in 2021, we did try some *Fungia* larvae in our isochoric chambers – and it did not work! Certainly timing and toxicity were probably issues.

However, we emphasize that our overarching motivation for all of this work is the development of real, field-deployable conservation and restoration tools, and whole coral fragments, given their ease of harvest and year-round availability, provide a much better biological model for these goals. We comment to this effect in the introduction and conclusions of the manuscript text.

5) It seems with the volume of the chamber multiple pieces of fragments can be loaded. Has this been tested?

Excellent observation – we did think about this early on, but decided to hold on varying sample size or amount until we had robust data on single fragment preservation, lest this should introduce an unanticipatedly complex new variable. This is because we wanted to ensure that we understood rigorously the parameters of the isochoric process during cooling and warming and the parameters of our husbandry process to get them to live and thrive once again after thawing in our seawater system with one small piece of coral. For restoration and thermodynamic purposes, in the future we plan to test a new sample sizes by keeping the cross-sectional area of the chambers largely the same but increasing their length to accommodate a longer piece of coral (0.5 x 11 cm²) instead of the current fragment size (0.5 x 0.5 cm²). If successful, these longer pieces could be cut post-thaw into several smaller pieces for grow-out to produce multiple adult corals.

6) A fundamental assumption for the usefulness of this novel technology is that coral can be restored or reestablished in the wild with microfragments after being bleached as described in the manuscript. Has this been done? How many species have demonstrated the feasibility and how easy is the process? This needs to be further explained and more references are needed.

We thank the reviewer for this question. Indeed reef restoration from microfragments is a cornerstone of modern coral conservation and restoration efforts, and the effects of short term bleaching are similarly well understood. We've added the following text to the Supplementary Information (and an according callout in the revised main text) to provide the reader more background on the subject, with several new references:

“Note S7: “Reef restoration from microfragmented coral

Restoration from microfragments grown to larger size and placed back on the reef is commonly done around the world by multiple coral restoration practitioners¹⁵. In particular, this process is so well-understood and widely accepted that the National Oceanic and Atmospheric Agency has developed a massive restoration process, called Mission Iconic Reef, to repopulate much of the Florida Reef Tract with the dominant reef-forming coral, *Acropora palmata*, strictly from microfragments (<https://sanctuaries.noaa.gov/news/dec19/noaa-launches-mission-iconic-reefs-to-save-florida-keys-coral-reefs.html>). The microfragmentation process is relatively easy, and tens to hundreds of species of coral throughout the world have been successfully microfragmented and returned to the wild either into in-water nurseries or onto the reef. Bleached corals, after the thermal or chemical stressors driving bleaching have ended, readily take up symbionts from their environment, as observed throughout nature when coral recover from bleaching in the wild¹⁷ and observed in the laboratory upon direct re-infection with symbionts^{16,18}. As stated in the conclusion of the main text, we thus anticipate that if future husbandry and stress modulation efforts can establish recovery of cryopreserved coral to the point of resumed calcification and growth, direct reef restoration from cryopreserved microfragments may present a promising route forward.”

References:

15. Page, C. A., Muller, E. M. & Vaughan, D. E. Microfragmenting for the successful restoration of slow growing massive corals. *Ecol. Eng.* **123**, 86–94 (2018).
16. Lager, C. *et al.* Metrics of Coral Microfragment Viability. *bioRxiv* (2023) doi:<https://doi.org/10.1101/2023.01.03.522625>.
17. Ritson-Williams, R. & Gates, R. D. Coral community resilience to successive years of bleaching in Kāneʻohe Bay, Hawaiʻi. *Coral Reefs* **39**, 757–769 (2020).
18. Wang, J.-T., Chen, Y.-Y., Tew, K. S., Meng, P.-J. & Chen, C. A. Physiological and Biochemical Performances of Menthol-Induced Aposymbiotic Corals. *PLoS One* **7**, e46406 (2012).

Minor:

Lines 63-67: It was stated that ‘However, while preserved reproductive materials can be utilized for reef restoration and diversification, climate change and its related bleaching events are negatively impacting coral reproduction. Therefore, a climate change-resistant approach is needed to enable cryopreservation of the entire coral organism without waiting for increasingly uncertain yearly reproductive events.’ Please specify how climate change affects coral reproduction and the application of cryopreservation for sperm and larvae. Based on the description, it is assumed that the collection of gamete samples and larvae can be very sensitive to climate change. If this is the case, it needs to be pointed out more clearly. Also, a ‘climate change-resistant approach’ is overstated. The present work introduced an alternative strategy for coral cryopreservation, but nothing is climate change-resistant. The process of sample collection and eventually restoration still needs to deal with climate change with this novel approach.

We thank the reviewer for flagging this, and we’ve accordingly revised this section of the text to be more clear and specific. It now reads:

“However, while preserved reproductive materials can be utilized for reef restoration and diversification, climate change and its related bleaching events are negatively impacting coral reproduction. Specifically, certain species, when exposed to warming waters and the stress of subsequent bleaching, demonstrate a long-term loss of sperm motility, reduction of egg size, and abnormal larval development (Hagedorn and Carter, 2016) or loss of reproduction the following years (Levitan et al. 2014). It is hypothesized that this multi-year impact may be due to ultraviolet radiation (UVR) damage incurred by reproductive stem cells during bleaching, when the coral loses the UVR protection produced by its symbionts (Henley et al. 2022). Therefore, an approach that can circumvent these climate change-related reproduction issues is needed, enabling cryopreservation of the entire coral organism without waiting for increasingly uncertain yearly reproductive events.”

References:

Hagedorn Mary, Carter Virginia L., Lager Claire, Camperio Ciani Julio F., Dygert Alison N., Schleiger Reuben D., Henley E. Michael (2016) Potential bleaching effects on coral reproduction. *Reproduction, Fertility and Development* **28**, 1061-1071.
<https://doi.org/10.1071/RD15526>

Levitan, Don R., Boudreau, William, Jara, Javier, and Knowlton, Nancy. 2014. "Long-term reduced spawning in *Orbicella* coral species due to temperature stress." *Marine Ecology Progress Series*. 515:1–10. <https://doi.org/10.3354/meps11063>

Henley EM, Quinn M, Bouwmeester J, Daly J, Lager C, Zuchowicz N, Bailey DW, Hagedorn M. Contrasting reproductive strategies of two Hawaiian Montipora corals. *Sci Rep*. 2022 Jul 18;12(1):12255. doi: 10.1038/s41598-022-16032-6. PMID: 35851072; PMCID: PMC9293913.

Lines 106-108: Need a reference here.

Added!

Reference:

Fahy, G. M. & Wowk, B. Principles of Cryopreservation by Vitrification. in 21–82 (2015). doi:10.1007/978-1-4939-2193-5_2.

Line 204: What is the definition of 'recover fully'?

We defined coral as being fully recovered when they have a full and functional compliment of their symbionts as measured by Pulse Amplitude Fluorometry and a colorimetric chart, as well as having a health metric evaluation similar to control corals. Because this was such an important issue, we wrote a paper on acceptable metrics and ways to evaluate healthy coral microfragments (Lager et al., 2023), which is referenced throughout the current work.

We have clarified in the text that this is the recovery standard that we are applying, thank you for flagging. New text below:

"Full recovery is here evaluated according to the standardized health metrics provided by Lager et al [Lager et al. 2023]."

References:

[Lager et al., 2023]

Metrics of Coral Microfragment Viability

Claire Lager, Riley Perry, Jonathan Daly, Christopher Page, Mindy Mizobe, Jessica Bouwmeester, Anthony N. Consiglio, Matthew J. Powell-Palm, Mary Hagedorn
bioRxiv 2023.01.03.522625; doi: <https://doi.org/10.1101/2023.01.03.522625>

Figures 3 and 4: Describe the hr post-thaw for all data presented.

We have clarified the description of the data presented in both of these figures, and added clarifications to the labeling both in the main text and the figure captions. Revisions highlighted in the new text.

Reference: Thorough proofreading of the reference list is needed. There are many errors (e.g., species name not italicized)

Thanks for the catch – fixed!

Reviewer #3:

Key Results

Powell-Palm et al. developed and tested a methodology to cryopreserve small microfragments of a living coral colony. They used a pressure sensor device embedded within the cryopreservation equipment to show when vitrification occurred successfully, tested a series of methods within two components of the cryopreservation process to optimize success, and then showed that cryopreserved specimens were successfully revived using respirometry. Overall, this is an incredible breakthrough for cryopreservation science and coral conservation.

We thank the reviewer sincerely for taking the time to review our work, and are warmed by their enthusiasm for the results (which we share!).

Validity

Although I had a hard time understanding some of the initial validation sections of the paper, the authors did a nice job using language that can be relatively easy to digest. The step by step figures aided in the understanding of the validation of vitrification process.

There could be more introduction into why little to no ice formation is critical to success (Page 2 lines 73/73) and exactly what vitrification is within the context of coral cryopreservation.

We thank the reviewer for flagging this, because successful translation of this technique to field applications in coral conservation require high digestibility and clarity. We have added additional language clarifying why minimal ice formation is critical, and expanded our discussion on the meaning and significance of coral vitrification throughout.

The respiration technique was a nice way to show that the corals were alive and functioning, but the increase in oxygen consumption associated with bacterial blooms in dying corals was difficult to digest a bit without a reference or comparison data (Page 8, line 270). These statements suggest a rapid increase in oxygen consumption when a coral dies, but bleached controls do not show this...but the injured cryo frags do. Perhaps that is what should be emphasize more or maybe I am just missing something.

We thank the reviewer again for motivating us to expand our discussion of this section. The reviewer has it precisely right – the “bacterial blooms” (wonderful phrase!) in dying corals do drive a significant increase in oxygen consumption. The point of confusion here might rest on the bleached corals – these are entirely healthy corals that have simply been stripped of their algal symbionts prior to cryopreservation. So within the context of Figure 3 and our broader results, the bleached corals can be thought of as the “healthy” or “positive” control, to be contrasted with the cryo-injured or negative control. Due to the bacterial bloom, the cryo-injured controls show a marked increase in oxygen consumption, as noted; the bleached coral, being healthy, do not. In order to help clarify this for future readers we have emphasized that the corals that have undergone the bleaching process are perfectly healthy, and that the data marked “Bleached” represents the healthy/positive control. Thank you for flagging!

Data and Methodology

All seems available within the text and supplemental documents

Analytical Approach

All analyses appear appropriate

Suggested Improvements

See comments within other sections of the review.

Clarity and Context

Page 2, line 60: change to 'sperm has been banked' - Done!

Page 2, line 83: change to 'warming, and advances in perfusion...' - Done!

Page 3, line 109: what is 'ice-Ih'?

Ice-Ih is the standard phase of ice most often encountered in nature, which has a hexagonal structure (hence the 'h') and is less dense than water. This nomenclatural distinction is made in cryogenic thermodynamics studies only because there are 17+ other phases of ice that become possible at ultra low temperatures and high pressures.

Page 3, line 114: change to 'in theory and experimentally...' - Done!

Figure 1. I couldn't quite figure out how the ice nucleation temperature was determined (~-30C for c and ~-80C for d) -

The detection marker for ice growth is the increase of the pressure – so if one identifies the point in time at which the pressure increase begins and then looks at the corresponding temperature, that gives the identified nucleation temperature.

Page 6, 'Selection and fragmentation' section should be written in past tense - Done!

Figure 2 – just wanted to give a shout out that this is a great figure - Thanks!

Page 7, line 221: identify what DMSO and PG are within the text - Done!

It seems that there are two variables that are experimentally tested: 1 is the diffusion time of the solution and I can't really figure out what the other variable is. Explain more. Is the diffusion time the 'tox' in Figure 3? If so, what exactly is the 'iso' variable. I am having a hard time deciphering this part.

We thank the reviewer for flagging, and have expanded the figure caption to make this point crystal clear. The "tox" data represent toxicity trials wherein we **only** expose the coral to the CPA loading/diffusion routine (for one of three possible diffusion time periods), but do **not** then attempt to cryopreserve them. The "iso" data represent the isochoric vitrification trials wherein

we first expose them to the same three CPA loading/diffusion times, but then seal them in the isochoric chamber and attempt to vitrify them, completing the cycle shown in Figure 2.

May want to check that the red and green blocks and lines within the Figures are appropriately accessible for colorblind readers.

Thanks again for a very thoughtful comment – we ran these colors by the graphic designer helping us out, who said he chose these shades to not cause any color blindness problems (which indeed plague other shades of green and red).

REVIEWERS' COMMENTS

Reviewer #1 (Remarks to the Author):

The authors have satisfactorily responded to all my questions and concerns in great detail and also satisfactorily dealt with the points raised by the other reviewers.

Reviewer #2 (Remarks to the Author):

Thanks for carefully revising the manuscripts. All my concerns have been addressed.

Reviewer #3 (Remarks to the Author):

All of my comments/suggestions have been appropriately addressed.